# Interfacial friction enabling ≤ 20 μm thin free-standing lithium strips for lithium metal batteries

Shaozhen Huang [1,4], Zhibin Wu [1,4], Bernt Johannessen [2], Kecheng Long[1], Piao Qing[1], Pan He[1], Xiaobo Ji[1,3], Weifeng Wei [1], Yuejiao Chen[1] & Libao Chen [1] ✉

A practical high-specific-energy Li metal battery requires thin (≤20 μm) and free-standing Li metal anodes, but the low melting point and strong diffusion creep of lithium metal impede their scalable processing towards thin-thickness and free-standing architecture. In this paper, thin (5 to 50 μm) and free-standing lithium strips were achieved by mechanical rolling, which is determined by the in situ tribochemical reaction between lithium and zinc dialkyldithiophosphate (ZDDP). A friction-induced organic/inorganic hybrid interface (~450 nm) was formed on Li with an ultra-high hardness (0.84 GPa) and Young's modulus (25.90 GPa), which not only enables the scalable process mechanics of thin lithium strips but also facilitates dendrite-free lithium metal anodes by inhibiting dendrite growth. The rolled lithium anode exhibits a prolonged cycle lifespan and high-rate cycle stability (in excess of more than 1700 cycles even at 18.0 mA cm$^{-2}$ and 1.5 mA cm$^{-2}$ at 25 °C). Meanwhile, the LiFePO$_4$ (with single-sided load 10 mg/cm$^2$) ||Li@ZDDP full cell can last over 350 cycles with a high-capacity retention of 82% after the formation cycles at 5 C (1 C = 170 mA/g) and 25 °C. This work provides a scalable approach concerning tribology design for producing practical thin free-standing lithium metal anodes.

Lithium-ion batteries (LIBs) have been extensively employed in consumer electronics, electric cars, and grid-scale energy storage systems. To satisfy the increasing demand from the market, there is an urgent need to break the bottleneck of the specific energy of LIBs by using advanced electrode materials[1–4]. Lithium (Li) metal is regarded as the ultimate high-energy-density anode material with a high theoretical specific capacity of 3860 mAh g$^{-1}$ and extremely low reduction potential of −3.04 V, which can be promising to replace the traditional graphite anode used in commercial LIBs whose theoretical specific capacity is only 372 mAh g$^{-1}$ [5–7]. However, lithium metal anode unfortunately suffers from volume fluctuations, dendrite growth, and side reactions with electrolytes[3,4,8–10].

Progress has been made in the stabilization of lithium metal anodes, but some practical challenges need to be tackled when the lithium anode matches with commercial cathodes. In practical applications, the commercial cathode usually matches an area capacity of 3–10 mAh cm$^{-2}$ of a lithium anode, which means that the thickness of the lithium metal only needs to be controlled within 15–50 μm[11–13]. Li shows strong diffusion creep even at room temperature due to its low melting point. Given the resulting viscosity and poor machinability,

[1]State Key Laboratory of Powder Metallurgy, Central South University, Changsha, Hunan, China. [2]Australian Synchrotron, ANSTO, Clayton, VIC 3168, Australia. [3]College of Chemistry and Chemical Engineering, Central South University, Changsha, Hunan, China. [4]These authors contributed equally: Shaozhen Huang, Zhibin Wu. ✉e-mail: lbchen@csu.edu.cn

large-scale production of thin lithium metal electrodes has been pro-ven problematic[13–16]. Extrusion-based production processes now use substrate lamination, with single-sided load thicknesses ranging from 20 to a few hundred μm, but this results in higher cost of production. Traditional methods, such as electrochemical deposition and molten lithium irrigation, show the high costs and risks for preparing thin lithium strips, which makes it difficult to achieve large-scale industrialization[12,17,18]. A new design is urgently needed to achieve ultrathin processing and preparation, as well as performance improvement at high current densities of lithium metal[19].

Bare lithium is challenging to process into thin strips owing to its high viscosity and low strength[14–16]. Recently, Cui et al. designed an ultrathin composite lithium strip by using a graphene oxide host[12]. Also, Liangbing Hu et al. developed a new method to prepare ultrathin Li-Sn anodes using a molten metal solution[11]. However, their approa-ches are complex and thus leaves room for further innovation. Theo-retically, creating a surface layer with high toughness and non-stickiness on lithium can be a promising strategy to engineer ultrathin lithium metal. As a classical metal engineering method, mechanical rolling can be promising for engineering the thickness of lithium metal at a low cost. More importantly, it is able to trigger unique tribo-chemical reactions on the surface of metal strips to form a robust tribochemical film with the help of functional lubricant oils[20]. There-fore, it can be developed as a new method to construct artificial solid electrolyte interface (SEI) for lithium metal anodes different from the conventional approaches such as spin coating, immersion and spray coating. In previous studies, anti-pressure additives and anti-wear additives are widely used in lubricant oils to improve processing effi-ciency and reduce friction, which helps enhance surface strength, decreases surface wear, and aids to separate the two surfaces in pro-cessing areas of steel or Al- and Mg-based alloys[21,22]. We are not aware that this technique has been applied to the field of lithium metal processing. Zinc dialkyldithiophosphate (ZDDP), as a traditional anti-pressure anti-wear additive, can decompose at rubbing interfaces to form a protective surface friction film[20,23–25]. Intriguingly, it should be possible to utilize the in situ tribochemical reaction generating the friction film to process ultrathin lithium strips, as it can act as both an electrochemical interface and a surface processing layer for the lithium metal anode. To our knowledge, there are no reports preparing ultrathin and high-performance lithium metal anodes by designing robust tribochemical films based on the mechanical rolling method.

In this work, we used the anti-pressure anti-wear additive, ZDDP, to engineer multifunctional interfaces on the lithium metal strips, which are derived from the tribochemical reaction between ZDDP and Li in the rolling process. Multiple crucial aspects are achieved for the ultrathin lithium strips; (1) high interfacial hardness, (2) suppression of lithium dendrite growth, (3) directional deposition of lithium by repeated plating/stripping and (4) the notable desolvation effect to realize the faster plating/stripping of Li anode at high current density. The thin lithium strips can be fabricated with controllable thicknesses ranging from 5 to 50 μm (1 to 10 mAh cm$^{-2}$), revealing much higher mechanical strength and better electrochemical performances than that of bare Li strips. The Li@ZDDP||Li@ZDDP symmetrical cells show ultralong cycling stability of up to 2800 h at 1.5 mA cm$^{-2}$ and 1.5 mAh cm$^{-2}$ while running more than 1700 cycles even at 18 mA cm$^{-2}$ and 1.5 mAh cm$^{-2}$, which is superior to Li||Li symmetrical cells. Sur-prisingly, it is also possible to maintain a cycle life of up to 2800 h even at a high area capacity of 5.0 mAh cm$^{-2}$ under 5.0 mA cm$^{-2}$. Besides, Li@ZDDP||Li@ZDDP symmetrical cell based on ultrathin lithium strips with 15 μm thickness can last for more than 800 h at 1.0 mA cm$^{-2}$ and 1.0 mAh cm$^{-2}$. The LiFePO$_4$(LFP)||Li@ZDDP full cell (the mass loading of LFP ~ 10 mg cm$^{-2}$) exhibits excellent cycling life with more than 83.2% capacity retention after 350 cycles, whereas the LFP||Li cell degrades rapidly. The outstanding electrochemical characteristics of the Li@ZDDP anode can be explained by the lithiophilic high-strength artificial SEI layer (surface hardness of 0.84 GPa, surface Young's modulus of 22.667 GPa) which inhibiting dendrite growth, inducing uniform Li plating and stripping, and alleviating the drastic interface fluctuation of the lithium anode. This work opens up a new strategy for scalable production of high-performance thin free-standing Li anodes.

## Results and discussions

As shown in Fig. 1a, thin lithium strips can be prepared by rolling at 25 °C in the Ar-filled glovebox with the addition of the oil mixture 5% ZDDP in mineral oil. Through a stepwise thinning process, the thick-ness of lithium strips can be controlled within a range from 50 μm to 5 μm (Figure S1, Supplementary Information). Moreover, the ZDDP can react with Li metals to form an organic/inorganic hybrid interface nanolayer on the lithium strip when loaded with high stress (Fig. 1b–f). Before processing, the chemical bonding information of pure ZDDP reagent was tested by infrared spectroscopy (Figure S2, Supplemen-tary Information), with strong P-O-C vibration observed in the spectral region of 920 to 1200 cm$^{-1}$ [26]. The high chemical activity of Li itself could generate an anti-pressure anti-wear film by high stress loading at room temperature. The contact pressure can induce catalytic decomposition of ZDDP[27]. At the same time, the infiltration of sulfur occurred. The cross-section profile of Li@ZDDP was observed by cryo-TEM, whose corresponding elemental mapping of P, S, and Zn was plotted in Fig. 1d. It can be observed that the in situ formed organic/inorganic interface layer has a thickness of approximately 450 nm, which could be further semi-quantitatively determined by time-of-flight secondary ion mass spectrometry (TOF-SIMS) in Fig. 1e. The top layer is an organic zinc polyphosphate (PH2O- atomic groups refer to organophosphate functional groups where H indicates C-H bond). It is inferred that the thickness of the layer of organic component in the anti-pressure anti-wear film is about 200 nm. In contrast, there is no corresponding components concerning Zn, P and S elements on the surface of bare lithium metal (Figure S3, Supplementary Information).

To further determine the components of the hybrid interface after reacting, X-ray photoelectron spectroscopy (XPS) was used to analyze its chemical structure and electronic information. Combined with the peak strength change, it is clearly found that the P element reaction layer was mainly within 100 nm of the surface (Fig. 2a). The distribution of Li has a trend of increasing with the depth (Fig. 2b). By detailed analysis of the *P 2p* spectrum, the surface of the layer interface mainly consists of organic polyphosphate. With increased sputtering time, the peak located at 133.8 eV, which can be assigned to organophosphorus components (-C-O-PO$_3$), gradually disappears[28]. Similarly, the *O 1s* spectrum can indicate that there are two peaks of 531.5 and 532.9 eV at the top layer, corresponding to the two chemical combinations of -O = P- and -O-C-, respectively (Fig. 2c). As sputtering time increases, the characteristic peaks of organic polyphosphates disappear and a new peak ~ 534.0 eV merges, which can be assigned to Li$_2$CO$_3$[29]. To further elucidate this change, the *C 1s* spectrogram was analyzed (Fig. 2c). There are a large number of organic C chains on the initial surface, and the -C-C- peak corresponds to 284.8 eV. This specific gravity decreases significantly as the depth increases, and significant C = O peaks of ~288.3 eV occur at depths of 100 nm and 200 nm. Based on this, it can be speculated that Li$_2$CO$_3$ is one of the main components in the inorganic layer. By analyzing the *S 2p* spectrogram, there are two signals appearing, S-M (M = Li, Zn) with 162.1 eV and S-P with 163.3 eV[30]. This indicates that S exists in two states: S-P and inorganic metal sul-fides. It has been confirmed that in the inorganic layer, there is mainly another inorganic substance − Li$_2$S. In conclusion, the surface post-reaction can be described as a layer dominated by an organic zinc polyphosphate layer, and the inward transition to a sulfur-rich layer dominated by Li$_2$S and P-S-Li. The inorganic compounds obtained by processing are main components of the SEI layer generated during the electrochemical cycle, which have faster Li$^+$ transfer rate[31,32]. XPS ana-lysis was also performed on the surface of the original lithium sheet,

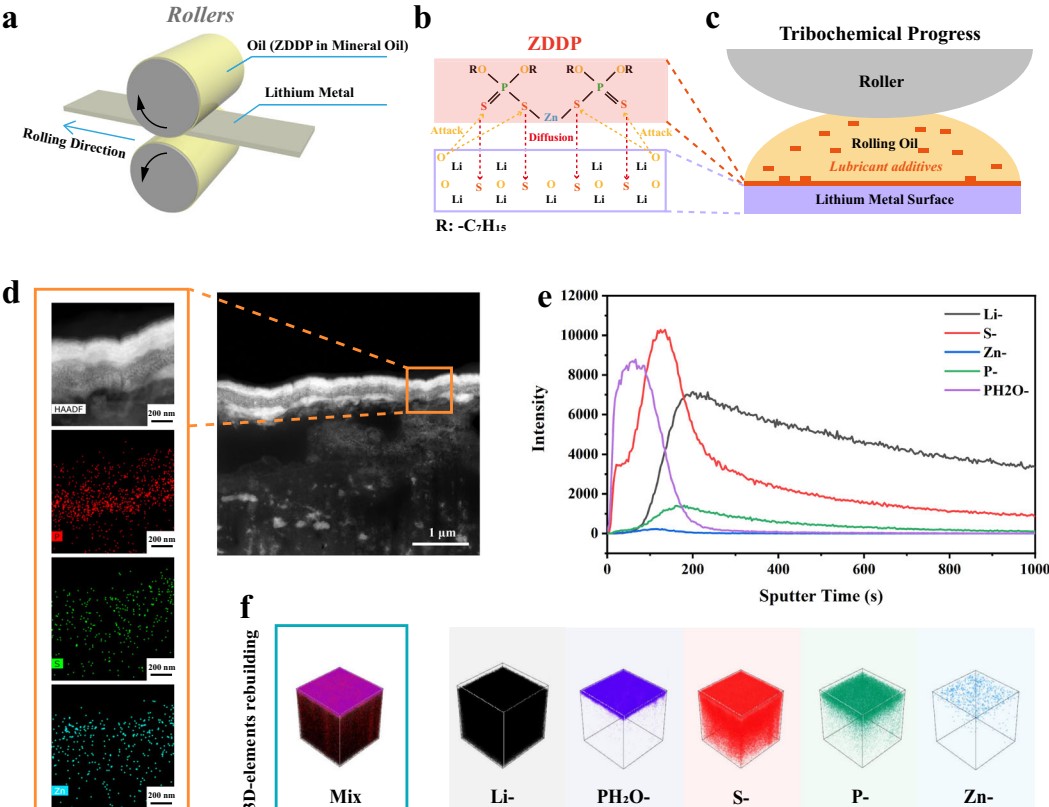

**Fig. 1 | Preparation of the Li@ZDDP strips and its formation mechanism analysis. a** Schematic diagram of the process of preparing lithium strip rolling using ZDDP additives. **b** Schematic of the mechanochemical reaction of ZDDP: ZDDP reacts under the catalysis of friction ansd alkali metal ions, and S diffuses to the lithium matrix and forms organophosphates. **c** Schematic diagram of the reaction of ZDDP molecules in oil droplets in the rolling process to form a film on the surface of lithium metal. **d** Photographs of the bifunctional nanofilm layer and elemental distribution generated by Li@ZDDP cross-sectional observation with cryo-TEM. **e** The TOF-SIMS profiles of different atom counts with the depth increasing on Li@ZDDP. **f** 3D structure views for TOF-SIMS depth sputtering on the surface of Li@ZDDP.

and $Li_2CO_3$, $Li_2O$ and $Li_3N$ were present, which are common products after Li contacting with dry air (Figure S4, Supplementary Information). The results suggest that the Li@ZDDP's (Figure S5, Supplementary Information) and bare Li's (Figure S6, Supplementary Information) basic inorganic components ($Li_2CO_3$, $Li_2O$) in the SEI layer are almost identical after cycles in the ester-based electrolyte (LB515, 1.0 M $LiPF_6$ in Ethylene carbonate (EC): ethyl methyl carbonate (EMC): Fluoroethylene carbonate (FEC) = 3:7:1 Vol%). The inorganic components– LiF mainly produce from $LiPF_6$ or FEC in the electrolyte. It is worth noticing that compared with the composition of bare lithium and Li@ZDDP after electrochemical reacting, Zn participated in the composition of the SEI of Li@ZDDP anode (Figure S5, Supplementary Information).

To elucidate the variation of electronic and local atomic environment around Zn element in Li@ZDDP, Cycled Li@ZDDP, and ZDDP, synchrotron X-ray absorption spectroscopy (XAS) experiments were conducted at the Zn K-edge at the Australian Synchrotron[33]. It is obvious that the valence states ($Zn^{2+}$) of the ZDDP was reduced upon the formation of the organic/inorganic hybrid interface nanolayer on the Li@ZDDP after rolling (Fig. 2e, f), while the first shell of ZDDP was changed from $ZnS_4$ tetrahedron to $ZnO_4$ tetrahedron (Fig. 2g, h, Figure S7, and Table S1, Supplementary Information). By fitting these Extended X-ray Absorption Fine Structure (EXAFS) of Li@ZDDP and ZDDP, we can obtain quantitative coordination information that the initial Zn-S bonds (2.33(1) Å) of the ZDDP molecule change to Zn-O bonds (2.01(2) Å) in the tribochemical film of Li@ZDDP, which gives solid evidence of a tribochemical reaction[34]. It is evident that the Zn–S

bonds of the ZDDP are decomposed during the rolling operation, resulting in Zn−O bonds instead, with the possible formation of zinc polyphosphates stated by other similar works[35,36]. However, LiZn alloys or lithium zinc polyphosphates may also be formed with zinc polyphosphates on Li@ZDDP in this work, as the valence state of Zn in Li@ZDDP is between $Zn^0$ and $Zn^{2+}$, and the intensity of the Zn-Zn bond at ~2.9 Å is relatively high. The local environment around Zn atom of Li@ZDDP before and after electrochemical cycling were also compared, indicating electrochemical stability of the organic/inorganic hybrid interface nanolayer on the Li@ZDDP (Fig. 2e, f). Minor differences between the XANES and EXAFS spectra of Li@ZDDP and Cycled Li@ZDDP can be ascribed to the lithium-ion migration which induces a subtle change to local order and stresses in the nanolayer[37]. In summary, a stable zinc polyphosphates interface nanolayers were formed on the surface of Li@ZDDP through the tribochemical reaction during the rolling operation.

This process layer formed by the ZDDP reaction has excellent mechanical properties, and the surface layer is mechanically tested by the nanoindentation measurements in continuous multi-cycles loading mode (schematic diagram shown in Fig. 3a). The results show that the surface Young's modulus of the processed lithium metal (Li@ZDDP) is about 25.90 GPa, which is about 3.5 times higher than the lithium strip (7.43 GPa, Fig. 3b). Crucially, the surface hardness is increased to 0.84 GPa (vs. 0.06 GPa of Li metal matrix), almost increasing to 15 times as compared with that of Li (Fig. 3c). For thin lithium strips, the adhesion wear rate can be significantly reduced with increased surface hardness[38]. As lithium strips become thinner, the

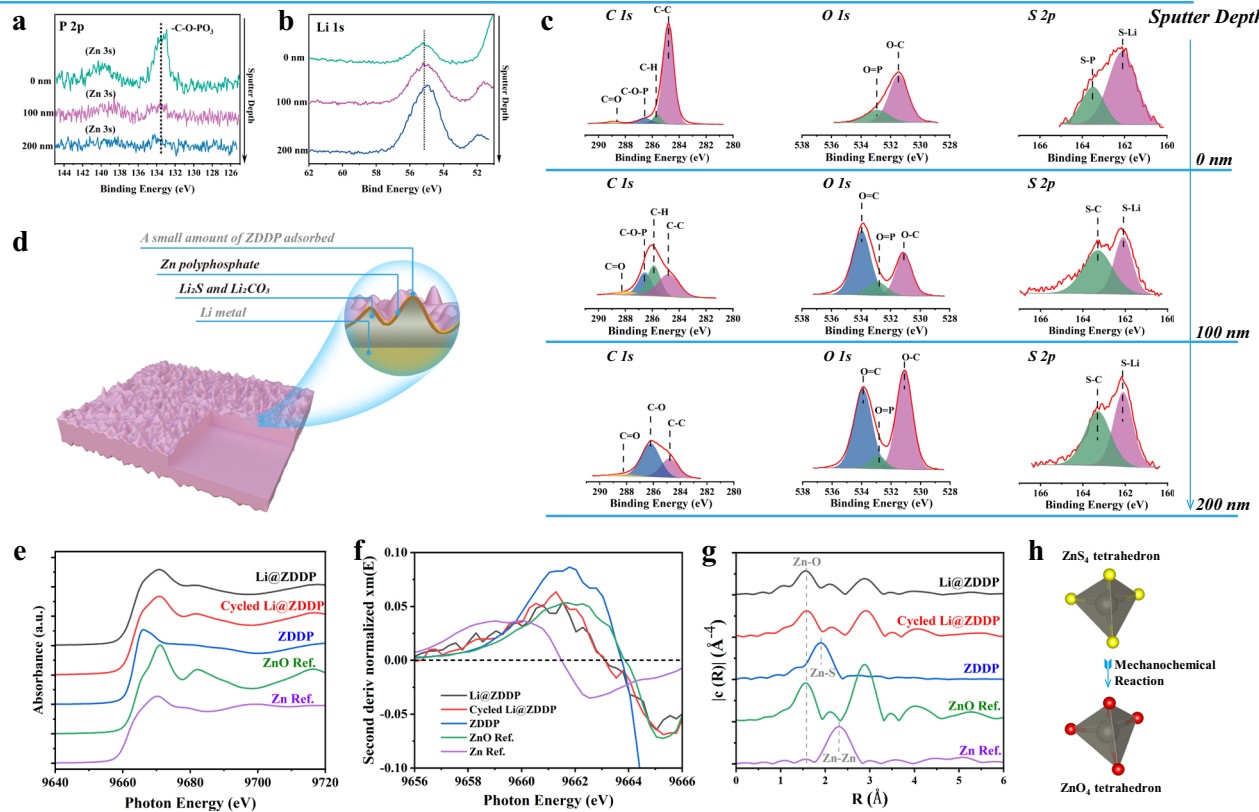

**Fig. 2 | Components and local structure of the organic/inorganic hybrid interface on Li@ZDDP. a** XPS depth profiling of *P 2p*. **b** XPS depth profiling of *O 1 s*. **c** XPS peak differentiation imitating analysis of *C 1 s*, *O 1 s* and *S 2p* in different sputtering depths of original Li@ZDDP. (**d**) Schematic model of the surface composition of the Li@ZDDP. **e** Zn K-edge X-ray absorption near edge structure (XANES) spectra, **f** Second order derivative of Zn K-edge XANES spectra, and **g** FTs of k3-weighted Zn K-edge EXAFS spectra of Li@ZDDP, Cycled Li@ZDDP (plating after 10 cycles at 1.5 mA cm$^{-2}$ and 1.5 mAh cm$^{-2}$), ZDDP, ZnO reference and Zn reference, respectively. **h** Local atomic structure changes around Zn before and after rolling for ZDDP.

ZDDP-derived harden surface would greatly improve the tensile strength due to the increased proportion of the surface layer relatively to the Li matrix. In summary, the high surface hardness of the tribofilm is conducive to separate the interface completely between the lithium strips and the roller when rolling toward thin scale. The entire reaction occurs only on the contacted surface in the rolling process and the high surface hardness of the nanoscale interface layer improves the overall mechanical processing properties. In order to verify the thin Li still having the strength, tensile performance tests were performed. The size of the strip tensile specimen of different thicknesses is shown in Fig. 3d. The specification is 25×4 mm. Since thin lithium fractures easily, rectangular specimens can be used for the tensile tests. The tensile results using these samples are shown in Fig. 3e. The improvement of mechanical properties can effectively improve the processing of thin lithium strips (Figure S8, Supplementary Information). With the same thickness of 50 microns, the tensile strength of bare lithium is 1.011 MPa and the Young's modulus is 103.336 MPa while the tensile strength of Li@ZDDP reaches 1.547 MPa and the Young's modulus is 250.181 MPa. When the thickness reduces to 15 μm, the tensile strength decreases to only 1.371 MPa, and the Young's modulus reaches 350.152 MPa. It can be seen that under the thin thickness, the Young's modulus is further improved mainly because the reduction of the thickness further amplifies the impact of the surface layer's mechanical strength. Numerous literatures have shown that the tensile curve has no obvious yield platform in bare lithium[14,15]. Overall, the increase in tensile strength is of great significance for realizing continuous rolling. Although exceeding the accuracy control of the rolling mill, it is able to prepare a lithium foil with a thickness of 5.40 microns (Fig. 3f), which means that the limiting conditions for thin

processing of lithium are no longer the material properties but the precision of the rolling mill. As a proof-of-concept, the rolled 15-μm thickness of lithium is highly flexible; it can be folded and rolled repeatedly and retains mechanical integrity, as shown in Fig. 3g.

The properties of the organic/inorganic hybrid interface have a significantly positive impact on the electrochemical properties of the material. At 1.5 mA cm$^{-2}$ and 1.5 mAh cm$^{-2}$, Li@ZDDP||Li@ZDDP symmetrical cell achieves a long lifespan up to 1400 cycles (Fig. 4a). Even at a high current density of 18.0 mA cm$^{-2}$, more than 1700 cycles can be achieved at Li@ZDDP||Li@ZDDP cell (Fig. 4b). In contrast, the polarization voltage of bare lithium significantly increases at less than 125 cycles at the same current density. At the test conditions of 5 mA cm$^{-2}$ and 5 mAh cm$^{-2}$, Li@ZDDP can also sustain a lifespan of more than 2800 h, while bare lithium fails after about 250 h (Figure S9, Supplementary Information). The Li@ZDDP at the 15 μm matching the 600 μm Li@ZDDP was assembled into a symmetrical cell. The 15 μm Li@ZDDP anode was plated first to examine the electrochemical performance of the thin anode. In this case, the cell still achieved a stable lifespan over 800 h (Figure S10, Supplementary Information). The constant current discharge tests on different thicknesses of rolled lithium are shown in Figure S10 (Supplementary Information). The areal capacity of lithium is approximately in line with the relationship of 1 mA h cm$^{-2}$ per 5 μm thickness, which is consistent with published literature[11–13]. The performance of the LB515 ester-based electrolyte were also verified. At 0.5 mA cm$^{-2}$ and 1.0 mA cm$^{-2}$, Li@ZDDP have longer lifespan and lower polarization (Figure S11, Supplementary Information) and more generally. Li@ZDDP has better polarization stability in the variable rate cycle test (Figure S12, support information). Obviously, the polarization of bare lithium begins to increase

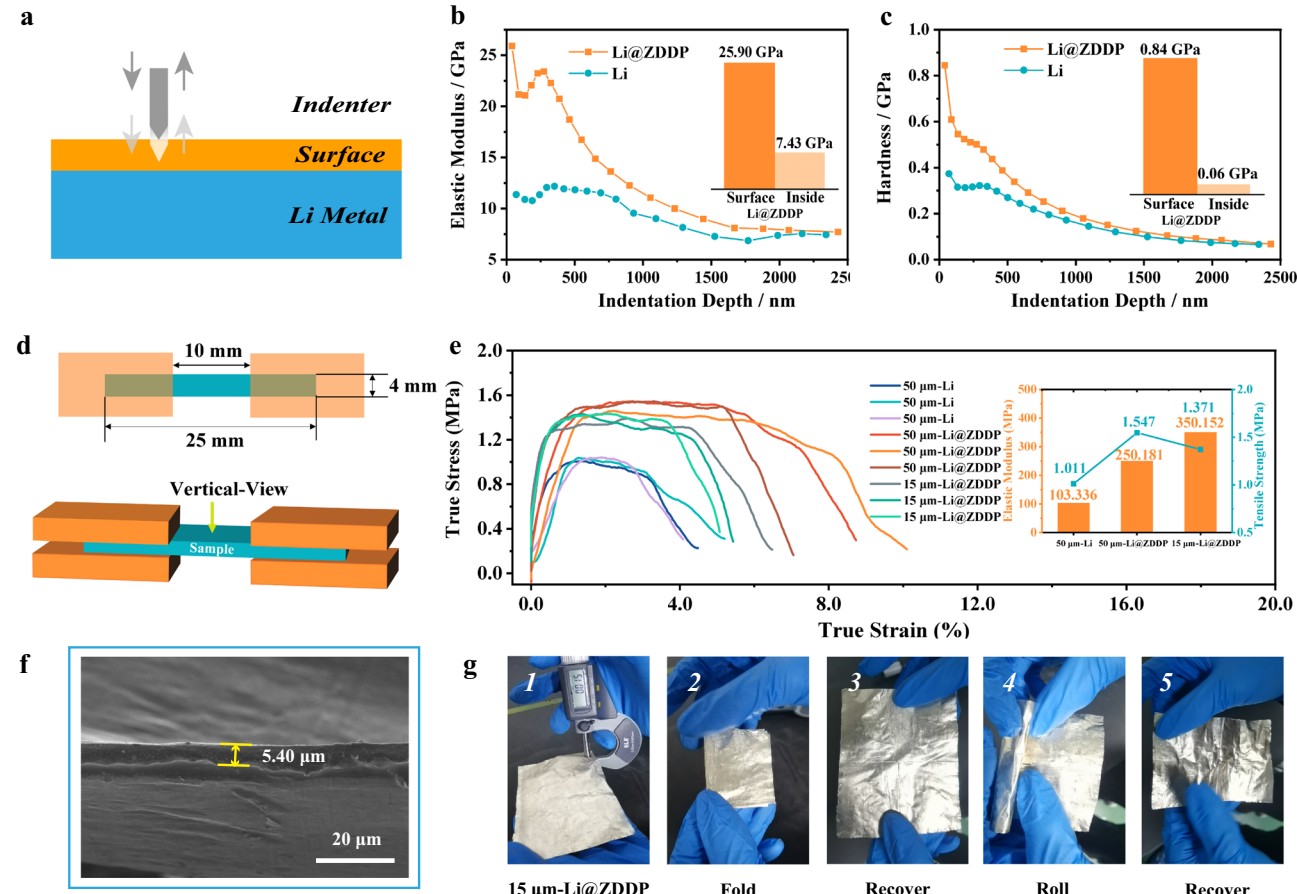

**Fig. 3 | Mechanical properties of Li@ZDDP strips. a** Schematic diagram of cyclic loading mode for nanoindentation test. **b** Comparisons of elastic modulus between bare Li metal and Li@ZDDP films using nanoindentation measurements. **c** Comparisons of hardness between bare Li metal and Li@ZDDP films using nanoindentation measurements. **d** Dimensional drawing of the stretched specimen. **e** True stress strain curves of Li and Li@ZDDP: 50 μm Li, 50 μm Li@ZDDP and 15 μm Li@ZDDP. **f** SEM image of the thinnest lithium foil (5-μm-thick) rolled exceeding the precision of the mill. **g** Digital camera image of centimeter-scale fabricated ultrathin Li@ZDDP foil (15-μm-thick) and abuse test.

when returning to the 0.5 mAh cm⁻². In order to understand the core reason of excellent cycle stability and high-rate performance on lithium anode, nanoindentation tests of electrodes after cycling, atomic force microscopy tests, SEM-EDS, and theoretical calculations were conducted. During the electrochemical cycle, the SEI layer of Li@ZDDP maintains significantly high strength (surface elastic modulus of 23.72 GPa vs. 8.62 GPa@Li, surface hardness of 0.64 GPa vs. 0.20 GPa@Li); a conclusion obtained from nanoindentation measurements (Figure S13, Supplementary Information). After 10 charge-discharge cycles at 1.5 mA cm⁻² and 1.5 mAh cm⁻², bare lithium and Li@ZDDP's SEI layer are measured by atomic force microscopy (Figure S13). Figures S13d and S13e show the cycle curves for loading and unloading of the probe. In the force curve spectrum, the SEI layer elastic modulus of Li (8.568 GPa) and Li@ZDDP (22.667 GPa) was obtained from the unloading curve. The elastic modulus of Li@ZDDP after cycling maintains the properties of the original surface, which also indicates that the reaction products of ZDDP and Li exist in the composition of the SEI layer. In the case of the SEI layer formation after the reaction, the adhesion force of Li@ZDDP decreases to 23% of bare lithium. It confirms that the nanoorganic/inorganic hybrid interface can effectively eliminate the viscosity of lithium, which is consistent with nanoindentation results. The loading and unloading curves of bare lithium SEI broadly overlaps, but at the same time, there is a significant viscous hysteresis in the unloading curve (*aob* region, Figure S13). This observation contrasts sharply with the response of the Li@ZDDP (Figure S13). There is a significant lag between loading and

unloading until the AFM tip completely detaches from the Li@ZDDP surface and returns to its normal position. There is no significant viscous hysteresis in the unloading curve. Under the single point load of the needle tip, the SEI layer of Li@ZDDP exhibits good deformation ability and a greater energy absorption effect locally. The local deformation increases the elastic modulus of the local SEI layer. At the same time, after obtaining the extra energy, the high Young's modulus of Li@ZDDP's SEI layer further reacts on the surface of the electrode, significantly increasing its hardness. This facilitates the continuous stabilization of the SEI layer during the cyclic process[39]. The SEI layer of Li@ZDDP has a higher elastic modulus, which helps maintain the flatness of the interface during the stripping/plating process[9,40]. This improvement in mechanical properties greatly improves the electrochemical properties of the lithium anode[41–43]. Li@ZDDP and bare lithium were stripped with different capacities at a current density of 1.5 mA cm⁻². Through the morphology of SEM images, the differences in stripping Li from Li@ZDDP with high-hardness surfaces are shown. The bare lithium interface creates potholes with uneven distribution and depth, whilst, on the other hand, Li@ZDDP exhibits a more uniform microporous Li stripping morphology (Figure S14, Supplementary Information). The Energy Dispersive Spectroscopy (EDS) images of Li@ZDDP after stripping showed that C, O, P, Zn have different degrees of morphological enrichment (Figure S15, Supplementary Information). It can be inferred that the surface organic layer plays a role in homogenizing the shedding of lithium ions. Further, SEM results show that bare lithium deposition is difficult to fill in the

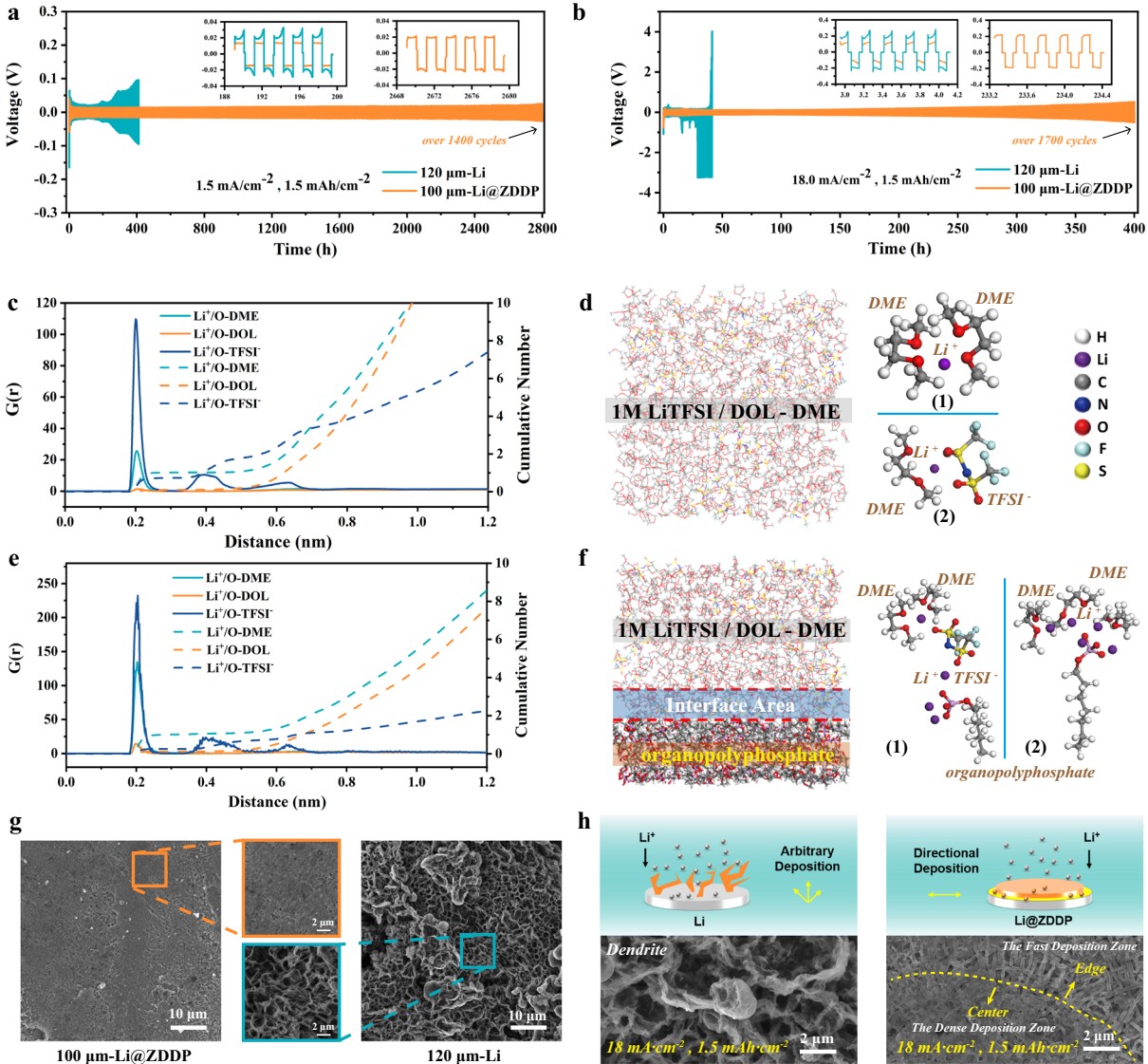

**Fig. 4 | Comparison of electrochemical behaviors of Li@ZDDP and bare Li. a** Lithium stripping/plating cycling of symmetric cells with Li@ZDDP (100-μm-thick) and bare Li (120-μm-thick) electrodes at 1.5 mA cm$^{-2}$ and 1.5 mA h cm$^{-2}$ using LS009 electrolyte. **b** Lithium stripping/plating cycling of symmetric cells with Li@ZDDP (100-μm-thick) and bare Li (120-μm-thick) electrodes at 18.0 mA cm$^{-2}$ and 1.5 mA h cm$^{-2}$ in the LS009 electrolyte. Molecular dynamics simulation **c–f: c** The radial distribution function g(r) (solid lines) and cumulative number (dotted lines) of the O atoms on DOL/DME molecules around Li-ions in 1 M LiTFSI/DOL-DME electrolyte. **d** Snapshots of the 1.0 M LiTFSI in DME:DOL = 1:1 Vol% solvation environment. **e** The radial distribution function g(r) (solid lines) and cumulative number (dotted lines) of the O atoms on DOL/DME molecules around Li-ions in the interface area. **f** Snapshots of the solution environment in the interface area. **g** SEM images of the deposition morphology between bare Li and Li@ZDDP electrodes after 100 cycles at 18.0 mA cm$^{-2}$ and 1.5 mA h cm$^{-2}$. **h** Schematic diagram of cyclic deposition mechanism at high current density.

previously generated potholes, which makes the electrochemical interface of bare lithium more undulating[44]. Drastic volume changes invariably triggers reconstitution of the SEI layer[45,46], and the consumption of electrolyte is also aggravated. Conversely, Li@ZDDP exhibits a superior surface recovery ability (Figure S16, Supplementary Information), which greatly inhibits the volume change of the lithium anode during cycling, thereby reducing the occurrence of side reactions.

To further study the reasons for the performance improvement at the condition of high current density, the interaction between the interface and electrolyte can be explored using molecular dynamics simulation. Through the integration of the radial distribution function g(r) (the dotted lines shown in Fig. 4c, e) we can map the number of atoms/molecules around the lithium ion. Comparing Fig. 4c, e, the existence of organophosphates in the interface area significantly reduces the number of TFSI$^-$ around lithium ions. In the electrolyte

system shown in Fig. 4d, the higher coordination number around lithium ions indicates that the electrostatic field of a Li$^+$ would get to the solvent molecules out of the first solvation shell, which is detrimental to Li ion deposition onto electrode interface[47]. Combined with Fig. 4d, f, it shows that organic products of Li@ZDDP anode effectively reduces the coordination number of lithium ions. In pure electrolyte system, there are two typical solvatized configurations under equilibrium structures (Fig. 4d (1)(2)). The coordination number of Li-O in both configurations is 4, but decreases to 3 (Fig. 4f (1)) and 2 (Fig. 4f (2)) at the interface area meaning that the interface after ZDDP reaction can effectively achieve lithium ion desolvation. This is of great help in enabling fast stripping/plating of lithium ions at the solid/liquid interface. To investigate the Li$^+$ solvation structure near the Li or Li@ZDDP interface, we conducted Raman spectral tests using 1.0 M LiTFSI in 1,2-dimethoxyethane (DME):1,3-Dioxolane (DOL) = 1:1 Vol% electrolyte. The initial Raman spectra results of bare Li and Li@ZDDP

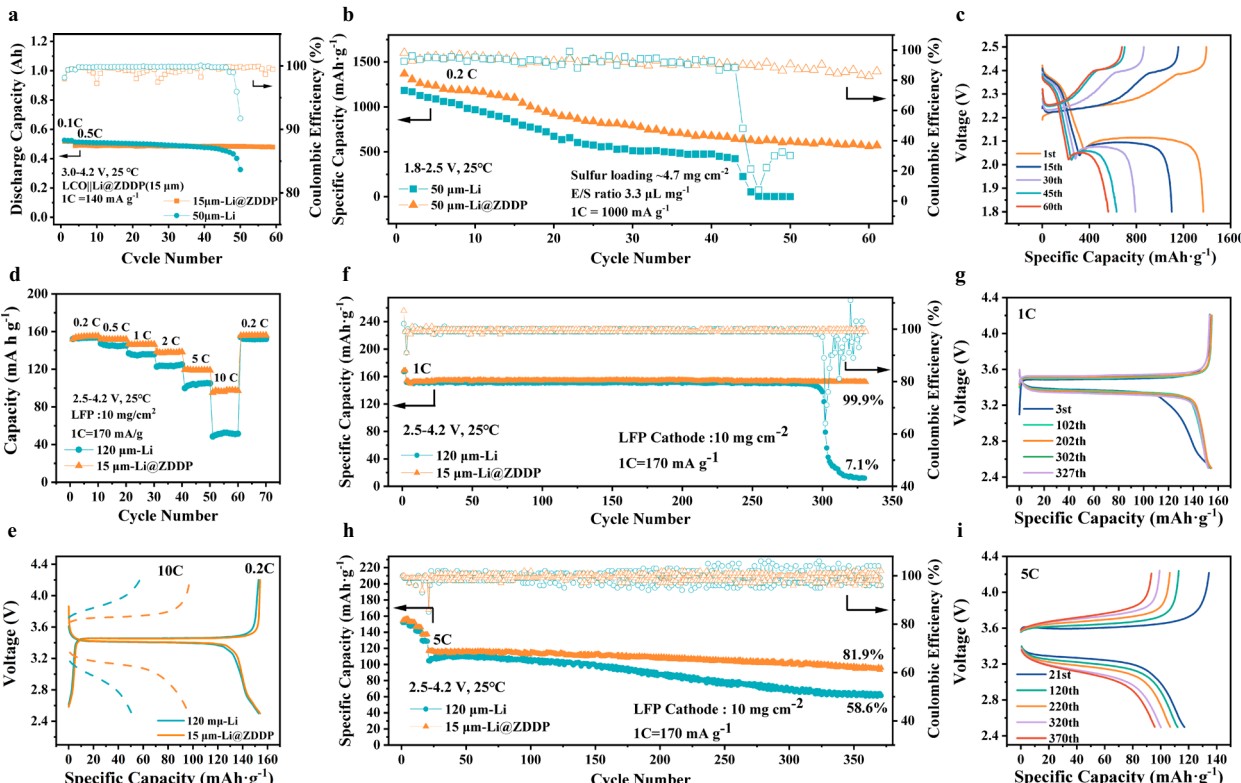

**Fig. 5 | Full batteries performances based on Li@ZDDP anodes. a** Cycling performance of the pouch cell with LiCoO$_2$||Li@ZDDP (15 μm) using 1.0 M LiPF$_6$ in EC:EMC:FEC = 3:7:1 vol% electrolytes at 0.5 C. **b** Cycling performance of the pouch cell with S||Li@ZDDP (50 μm) using 0.6 M LiTFSI in DME:DOL = 1:1 Vol% with 4.0% LiNO$_3$ electrolytes. **c** Galvanostatic charge/discharge profiles at different cycles using pouch cell with 50 μm anode. **d** Cycling performance of LFP||Li and LFP|| Li@ZDDP full cell at different rates. **e** The corresponding voltage profiles of LFP||Li and LFP||Li@ZDDP cells. **f** Cycling performance of the LFP||Li and LFP||Li@ZDDP full cells at 1 C rate. **g** The charge-discharge profiles of LFP||Li@ZDDP at 1 C rate after cycling. **h** Cycling performance of the LFP||Li and LFP||Li@ZDDP full cells at 5 C rate. (i) The charge-discharge profiles of LFP||Li@ZDDP at 5 C rate after cycling.

near the interface are shown in Figure S17c (Supplementary Information). The cation-anion clusters of Li@ZDDP-0 min (-736.8 cm$^{-1}$) assigned as free anion is different with that of LiTFSI (-755.1 cm$^{-1}$). Meanwhile, the cation-anion clusters of Li-0 min could be assigned as FA as well. With prolonged electroplating time, the S−N−S bending peak of the TFSI$^-$ at 736.80 cm$^{-1}$ (near the Li@ZDDP interface) undergoes blueshift, which indicates that the cation-anion clusters near the surface of Li@ZDDP transform to contact ion pair (CIP) and aggregates (AGG) states shown in Fig. 4f [48]. The peak of Li@ZDDP-14 min (point 8.0, 0–14 min labeled point 0.0–8.0) shown in Figure S17d (Supplementary Information) is about 747.0 cm$^{-1}$ while Li-14 min (point 8.0, shown in Figure S17e, Supplementary Information) still shows no distinct peak corresponding to free anion state. This indicates that Li@ZDDP can realize the desolvation effect when electroplating. Figure S17a and S17b (Supplementary Information) presents the in situ Raman data of Li@ZDDP and Li as a 2D contour plot with intensity in color covering a selected region between 600 and 1050 cm$^{-1}$, respectively, where the TFSI$^-$ peaks are framed. According to the principle of local "electrical neutrality", the dynamic change of TFSI$^-$ during electrodeposition can reflect the change of the concentration of Li$^+$. With the increase of the electroplating time, the concentration of Li$^+$ near the interface on Li is rapidly depleted due to the dendrite growth while the concentration of Li$^+$ has no obvious change near the Li@ZDDP due to the fast adsorption of the organic/inorganic hybrid interface and the continuous uniform deposition [49]. Consequently, the Li@ZDDP anodes exhibit better performance at larger current densities.

Furthermore, the interaction between organic SEI and lithium ions was studied by density functional theory (DFT) calculations, as shown in Figure S18 (Supplementary Information). The products of the

reaction (containing zinc organic lithium polyphosphate, lithium organic polyphosphate) and the external composition of the traditional SEI by organic species were compared (CH$_3$OCO$_2$Li and CH$_3$OLi in the ester-based electrolyte, CH$_3$OLi and CH$_3$CH$_2$OCH$_2$OLi in the ether-based electrolyte) [4,28,29,50]. The adsorption energy $\triangle E_{ads}$ is corresponded with $\triangle E_{ads} = E_{total} - E_{molecule} - E_{Li}$, where $E_{total}$ represents the total energy of the adsorbed Li species with catalyst molecule, $E_{molecule}$ is the energy of the empty organic molecules that do not bind lithium ions, and $E_{Li}$ is the energy of the Li-ion. Our results show that the adsorption energy of the obtained organic phosphonic acid (-519.58 kJ/mol) is much higher than ester-based SEI and ether-induced SEI. The presence of zinc can further reduce the adsorption energy (-540 kJ/mol). This is due to the interaction of the introduced zinc and the anion group, which further reduces the potential of the O site of adsorption of Li and obtains more activation sites. The electrostatic potential calculation results support this view (Figure S18, Supplementary Information). EIS for different cycles of the Li and Li@ZDDP electrodes were measured to reveal their interfacial evolution during high-current-density cycling as seen in Figure S19 (Supplementary Information). As illustrated in Figure Table S2 (Supplementary Information), Li@ZDDP displayed the smaller value of Li$^+$ transfer resistance ($R_{SEI}$) in both 0 and 1 cycle, indicating its fast ionic conductivity which was beneficial to the application at high current densities. The smaller charge transfer resistance ($R_{CT}$) was also reflected in the higher electrochemically activity. At high current density (18.0 mA cm$^{-2}$), Li$^+$ is deposited rapidly, and dendrite growth is unavoidable for bare lithium (with no surface modification). The SEM images of Li@ZDDP and Li after cycling are shown in Fig. 4h. A significant contrast can be observed at the edge of the deposition zone. Due to the more

lithiophilic sites and excellent mechanical properties of Li@ZDDP's interface, Li$^+$ can be densely deposited in the direction parallels to the surface[19,51]. The surface morphology of anodes after running 100 cycles is shown in Fig. 4g. The dendrite growth of bare lithium is significant and would severely restrict performance. On the other hand, Li@ZDDP has a more uniform and dense morphology. Further, it is determined that the mineral oil does not have a negative impact on the electrochemical cycling process (Figure S20, Supplementary Information). It was confirmed that an improved performance was brought on by ZDDP. Mineral oil does not significantly improve the electrochemical performance of lithium anodes. In summary, the organic layer provides more lithiophilic sites and stronger desolvation effects whilst the inorganic layer (Li$_2$S) exhibits a faster Li$^+$ transfer rate[32]. In combination, the organic/inorganic hybrid surface nanolayer effectively suppresses dendrite growth.

To verify the practical application of the prepared thin Li@ZDDP anode, full cells of S||Li@ZDDP, LFP||Li@ZDDP, LCO||Li@ZDDP were assembled, respectively. Figure 5a shows the cycling performance of the pouch cell using 15 μm Li@ZDDP (-3 mAh cm$^{-2}$) anodes and commercial LiCoO$_2$ cathodes with a double-sided load of 37.7 mg cm$^{-2}$ to evaluate its application performance. Its theoretical discharge capacity is about 0.56 Ah (-2.6 mAh cm$^{-2}$). The capacity retention of LCO||Li@ZDDP remains -100% after 50 cycles, while the capacity retention of LCO||Li keeps only 60% after 50 cycles, revealing that the Li@ZDDP anode exhibits better electrochemical stability. The capacity retentions were calculated regarding the formation of cycles at 0.5 C rate only. The performance of lithium-sulfur battery was shown in Fig. 5b. The electrolyte/sulfur (E/S) ratio was 3.3 μL mg$^{-1}$ to explore the performance of the thin anodes. With a large amount of Li$_2$S on the surface of Li@ZDDP anodes, the capacity of the pouch cell would be higher than that using bare Li for reducing the consumption of S from the cathodes. In lean liquid state, S||Li@ZDDP battery can still maintain stable cycling performance. Due to dendrite growth and sulfur corrosion, the lean electrolyte and bare Li anode with limited thickness were continuously reacted, which eventually leads to rapid failure in the S||Li cell. The trend of slowly decreasing discharge specific capacity during 60 cycles shown in Fig. 5c. A full cell configuration of LFP cathode with a load of 10 mg cm$^{-2}$ and an area capacity of nearly 2 mAh cm$^{-2}$ was applied under laboratory conditions. The 15 μm Li@ZDDP anode continues the high current characteristics tested in symmetrical batteries. LFP cathodes matched with thinner negative electrodes Li@ZDDP as full cell. The lithiophilic and desolvent interface with high modulus makes the lithium anode adapt to higher current applications. Discharge rate tested from 0.2 C to 10 C, the discharge specific capacity of Li@ZDDP is increasing to 2 times as compared with that of 120 μm Li anode (Fig. 5d). As shown in Fig. 5e, the discharge specific capacity of LFP||Li at 10 C rate is even less than 50% of LFP||Li@ZDDP. It can be confirmed that using the thinner lithium anode to assemble the full cell, the performance still exceeds the thicker lithium anode. 1 C cycling test was performed by LFP||Li@ZDDP full cell, achieving in excess of 325 cycles with a capacity retention rate of up to 99.9%. The capacity retentions were only calculated regarding the formation of cycles at 1 C rate only. On the other hand, the cell using bare lithium anode occurred short circuit due to dendrite growth, then causing rapid capacity decline (Fig. 5f). Figure 5g confirms that the discharge specific capacity does not decline after 325 cycles at the formation of 1 C rate cycles. Long cycle testing was performed at high rate (5 C). After 350 cycles, the capacity retention rate of the LFP||Li@ZDDP still exceeds 80% (Fig. 5h). The capacity retentions were just calculated regarding the formation of cycles at 5 C rate. This indicates that the constructed interface can both effectively inhibit the occurrence of side reactions and improve the cycle stability to a significant extent.

In summary, we have employed an interfacial friction-induced tribochemical reaction method to generate an organic/inorganic

hybrid surface nanolayer in situ on a lithium strip. This approach improves both the electrochemical properties and processing feasibility toward thin strips of lithium metal anodes. Through the rolling process, thin (5–50 μm) lithium anodes with controllable area capacities were achieved reliably and reproducibly. The ZDDP has an anti-pressure anti-wear effect, producing harder and higher-modulus nanolayers in the high-load large-deformation conditions. Li@ZDDP||Li@ZDDP is achieved over 1400 cycles not only at 1.5 mA cm$^{-2}$ and 1.5 mAh cm$^{-2}$ but also at 5 mA cm$^{-2}$ and 5 mAh cm$^{-2}$. Specifically, the Li@ZDDP anode performs over 1700 cycles at 18 mA cm$^{-2}$ and 1.5 mAh cm$^{-2}$. In LFP||15 μm-Li@ZDDP full battery testing with LFP (loaded with 10 mg cm$^{-2}$), the capacity retention rate exceeds 80% over 350 cycles at 5 C. Ultra-high modulus SEI layer (22.667 GPa) with organic/inorganic hybrid components which provide more lithiophilic sites and strong desolvation effect can realize the excellent electrochemical properties in high-current and high-volume applications. The same method is expected to be applied to a variety of lithium alloys, achieving large-scale high-performance thin-scale lithium metal anode preparation, and further improving the specific energy of existing batteries.

## Methods

### Fabrication of thin lithium anodes

The zinc dialkyldithiophosphate (ZDDP) was purchased from Guangdong Wengjiang Chemical Reagent Co., Ltd. Different thickness (100 μm, 50 μm, 30 μm, 15 μm) of Li was processed by using the lubricants (5 wt% ZDDP in mineral oil) to coat rollers of the mill (MRX-DG150L, Shenzhen Mingruixiang Automation Equipment Co., Ltd) based on 120 μm Li foils (99.9%, China Energy Lithium Co., Ltd, Tianjin) at 25 °C in the Ar-filled glovebox. The 50 μm bare Li used in lithium sulfur battery was purchased from China Energy Lithium Co., Ltd in Tianjin. The rolled lithium anodes with mineral oil (MO, Aladdin) were cleaned with tetrahydrofuran (THF, Dodo Chem) solution and then dried in the Ar-filled glovebox.

### Fabrication of cathodes

LiCoO$_2$ Cathodes. The cathode slurry comprising LiCoO$_2$, super P and polyvinylidene fluoride (PVDF) (All were purchased from Dongguan Large Electronics Co., Ltd. Guangdong, China) at the mass ratio of 94.9%:2.55%:2.55% were fully dispersed in N-methylpyrrolidone (NMP, Aladdin) for half an hour, then coated onto the 12 μm Al foil and dried overnight under vacuum at 120 °C. LiCoO$_2$ cathode had an active material loading of 37.7 mg cm$^{-2}$ (double sides) and the cathode thickness after rolling was 0.125 mm. LiFePO$_4$ Cathode. The cathode slurry comprising LiFePO$_4$, super P and polyvinylidene fluoride (PVDF) (All were purchased from Dongguan Large Electronics Co., Ltd. Guangdong, China) at the mass ratio of 94.7%:2.65%:2.65% were fully dispersed in N-methylpyrrolidone (NMP, Aladdin) for half an hour, then coated onto the 14 μm C@Al foil and dried overnight under vacuum at 120 °C. LiFePO$_4$ cathode had an active material loading of 10.0 mg cm$^{-2}$ (single side) and the cathode thickness after rolling was 0.085 mm. S/C Cathode. Sulfur powders (Alfa Aesar) and carbon nanotube (Zhongke era) (mass ratio = 6:4) were ground well with a mortar, then sealed the mixture in an Ar-filled ampoule and placed in a muffle furnace (Hefei Kejing) for 48 hours at 300 °C (S/C). The cathode slurry comprising S/C, super P and polyvinylidene fluoride (PVDF) at the mass ratio of 8:1:1 was fully dispersed in N-methylpyrrolidone (NMP, Aladdin) for half an hour, then coated onto the 12 μm Al foil and dried overnight under vacuum at 60 °C. S/C cathode had an active material loading of 4.7 mg cm$^{-2}$ (double sides) and the cathode thickness was 0.290 mm.

### Materials characterization

The thickness of the Li anode and surface morphologies of Li electrodes were characterized using thermal field emission SEM (MIRA4

LMH, TESCAN, Czech) operated at 5.0 kV. Test samples were prepared in a glovebox then transferred via sealed glass bottles with aluminum plastic bag packaging in vacuum. The XPS measurements were performed by Thermo Scientific K-Alpha$^+$ setup with a monochromatic Al Ka X-ray source and an Ar$^+$ sputtering gun (Thermo Fisher) from Shiyanjia Lab (www.shiyanjia.com). Test samples were prepared in a glovebox and quickly transfer into XPS measurements via sealed glass bottles with aluminum plastic bag packaging in vacuum. A cross-sectional sample of cryo-TEM was made by focused ion beams (FIB, Helios 5 UX). The surface layer thickness as well as the distribution of elements were observed by cryo-TEM (Spherical aberration correction field emission transmission electron microscope, Titan G2 60-300, FEI). Composition of surface was characterized by time-of-flight secondary ion mass spectrometry (TOF-SIMS, Gmhb 5, Münster, Germany) with an Ar-ion beam. The sputtering speed was about 1.5 nm s$^{-1}$. Bare Li, Li@ZDDP pre- and post-cycle surfaces were tested in the Ar-filled glovebox using the nanoindentation measurements (PI950, Bruker) under cycle loading mode. The tensile curves were obtained using a micro-tensile sample stage (Gatan MTEST2000) test under a scanning electron microscope (FEI Quanta 250 FEG). For post-cycle lithium and Li@ZDDP, we used atomic force microscopy (Bruker dimension icon AFM) in the Ar-filled glovebox to characterize surface SEI layer.

## In situ Raman characterization

Raman spectra tests were conducted with a Horiba Jobin Y von HR Evolution Raman spectrometer under a Raman laser wavenumber of 532 nm from a neodymium-doped yttrium aluminum garnet (Nd:YAG) laser operating at 120 mW. The Li deposition in Raman spectroscopic cell (K007, Tianjin Aida Hengsheng Technology Development Co., Ltd) with 1.0 M LiTFSI in DME:DOL = 1:1 vol% was performed at 3 mA cm$^{-2}$ lasting 20 min which implies a theoretical deposition thickness of ~5 μm, while the laser spot has a diameter of ~1.5 μm. For the in situ Raman test, the distance between the laser spot and the substrate was set as 6.5 μm.

## Synchrotron XAS characterization

Synchrotron XAS data was collected at the wiggler XAS Beamline (12ID) in Hutch B at the Australian Synchrotron (Melbourne, Australia). The experimental samples, including Li@ZDDP, Cycled Li@ZDDP (after cycling for 5 cycles), ZDDP, and ZnO powder, were measured in fluorescence mode at the Zn K-edge (9658.6 eV), with the measurement of a Zn reference foil simultaneously for initial beam energy calibration and proper energy alignment of multiple scans. Li@ZDDP, Cycled Li@ZDDP, ZDDP, and ZnO powder was mixed with cellulose, and then was pressed into round pellets (D = 7 mm) for use, respectively. The XAS data were aligned, deglitched, and normalized by using the Athena program with standard procedures. The built-in AUTOBK algorithm was adopted to limit background contributions below Rbkg = 1 Å. A k-weight of 3 was used to amplify the oscillations in upper end of k-space that is converted from the EXAFS spectrum. The coordination information can be obtained by Fourier transform (FT) of the EXAFS spectrum into the R-space. The main EXAFS parameters, including the coordination distance ($R + \Delta R$), coordination number (N), amplitude reduction factor (S02), Debye–Waller factor ($\sigma^2$), energy shift ($\Delta E_0$), and the R-factor, can be obtained by fitting by using the Artemis program. The Zn K-edge spectra of the samples were fitted with k, k2, and k3 weighting over a k-range from 2.7 to 11.0 Å$^{-1}$ and an R-range from 1.0 to 2.0 Å for Li@ZDDP and Cycled Li@ZDDP, and 1.0 to 2.3 Å for ZDDP. $S_0^2$ was fixed to 1.0 for Zn K-edge spectra based on the $S_0^2$ value obtained from fitting both the Zn foil reference and the ZnO reference. The error range of N was below 25%, and the uncertainty of coordination distance was below 0.02 Å, and all R-factors were controlled to be below 2.0%. The error bars were obtained from running the Artemis program with the EXAFS fitting.

## Electrochemical measurements

The discharge/charge testing were conducted on a Neware Battery system. All electrochemical tests were performed in a temperature-controlled room at 25 °C. All batteries employed Celgard®2400 as the separator. Lithium metal anode can theoretically deposit unlimited lithium ions, which is different from the anodes of the intercalation type such as graphite. Calculate the N/P ratio based on the amount of Li stripping from the anode. The measurement of the free Li capacity in thin electrodes was done by charging the battery to 1 V at 0.5 mA cm$^{-2}$ with the bare Li as the counter and reference electrode. Coin cells tests. CR2016 coin cells were assembled in an Ar-filled glovebox (O$_2$ and H$_2$O < 0.5 ppm). Symmetric coin cells (fresh lithium on each side or Li@ZDDP foil on each side) were assembled with 1.0 M LiTFSI in DME:DOL = 1:1 vol% with 2.0% LiNO$_3$ (LS009, Dodo Chem) as the electrolyte. To standardize the testing, 75 μL of electrolyte was used in each coin cell. Typically, we used a protocol of 1 h of stripping followed by 1 h of plating with a current density of 1.5 mA cm$^{-2}$ to achieve an areal capacity of 1.5 mAh cm$^{-2}$. The current density for the Li metal plating/stripping was set to 1.5 or 18.0 mA cm$^{-2}$. Besides, LFP||Li@ZDDP (N/P = 1.8) or LFP||Li (N/P = 14.1) full cells were assembled with 1.0 M LiPF$_6$ in EC:EMC:FEC = 3:7:1 vol% (LB515, Dodo Chem) as the electrolyte, which were tested within the voltage range of 2.5-4.2 V. As for cycling tests, the LFP full cells were activated at 0.1 C for 2 cycles firstly, then underwent the charge–discharge test at 1 C (1 C = 170 mA g$^{-1}$). The LFP full cells were activated at 0.1 C, 0.5 C, 1 C and 2 C (each rate for 5 cycles) firstly, then underwent the charge-discharge test at 5 C (1 C = 170 mA g$^{-1}$). As for rate performance tests, the LFP full cells underwent the galvanostatic charge-discharge test in turn of 0.2 C, 0.5 C, 1 C, 2 C, 5 C, 10 C and 0.2 C (1 C = 170 mA g$^{-1}$). Pouch cells tests. The LiCoO$_2$||Li@ZDDP (or Li) pouch cells and S||Li@ZDDP (or Li) pouch cells were assembled in dry rooms with dew points below - 40 °C. As for the LiCoO$_2$||Li@ZDDP (or Li) pouch cell, the size of the Li anode was 50 mm × 80 mm while the size of the LiCoO$_2$ cathode was 47 mm × 77 mm. Each electrode had the pole lug at the size of 8 mm × 12 mm. The width of the polypropylene separators was 83 mm. In the layer-by-layer manufacturing process, the LiCoO$_2$||Li@ZDDP (N/P = 1.1) or LiCoO$_2$||Li (N/P = 3.7) pouch cells used three cathodes and four anodes while lithium sulfur pouch cells used two cathodes (30 mm × 70 mm) and three anodes (33 mm × 73 mm). Aluminum electrode lugs were welded to the cathodes by an ultrasonic welder while nickel electrode lugs were welded to the anodes by mechanical pressing. LiCoO$_2$||Li@ZDDP (or Li) pouch cells were tested over the voltage range of 3.0-4.2 V. The pouch cell was activated at 0.1 C for 3 cycles firstly, then underwent the galvanostatic charge-discharge long cycle-lifespan test at 0.5 C (1 C = 140 mA g$^{-1}$). S||Li@ZDDP (N/P = 2.1) or S||Li (N/P = 2.1) pouch cells were tested over the voltage range of 1.8−2.5 V. Each pouch cell was tested at 0.2 C (1 C = 1000 mA g$^{-1}$). The electrolyte/sulfur (E/S) ratio was 3.3 μL mg$^{-1}$. Electrochemical impedance spectroscopy (EIS) tests were conducted on a Holland Ivium electrochemical station. The EIS data was recorded in the frequency range from 0.1 to 100,000 Hz.

## Computational methods

The adsorption energy and electrostatic potential calculation calculations have been carried out using the DMol3 program based on density functional theory (DFT) within GGA-PBE schame for exchange and correlation potential. We used DFT semicore pseudopotential with double numerical basis set plus polarization functions (DNP). Convergence in energy, force, and displacement was set as 10$^{-5}$ Ha, 0.001 Ha/Å, and 0.005 Å, respectively. The core treatment was set to effective core potentials (ECP). The space cutoff radius is maintained at 4.4 Å. Molecular dynamics (MD) simulation was performed using Gromacs (2019.5 version). For system one (the 1.0 M LiTFSI in DME:DOL = 1:1 vol% solvation environment), 250 DME, 350 DOL, 50 Li$^+$, and 50 TFSI$^-$ were added to 5 × 5 × 5 nm simulation box. For system two

(the interface area of the system two) 200 Li+ and 100 organophosphates were added to $5 \times 5 \times 1.5$ nm simulation box as the bottom layer and 250 DME, 350 DOL, 50 Li$^+$ and 50 TFSI$^-$ were added to $5 \times 5 \times 3.5$ nm top layer. The OPLS-AA force field with the charges refitted using RESP approach were adopted. For each system, the simulation started with a 2 ns NPT at 500 K and was followed by a 3 ns NPT at 330 K to make sure for the full dissolution of the electrolyte. Then, the systems were equilibrated at 298 K in the NPT ensemble for 5 ns and in the NVT ensemble for 10 ns. The last 5 ns trajectory is used to obtain the structure and the Radial distribution function (RDF) for the interface selected the regions that more than 1 nm away from the bottom layer.

## Data availability

The data that support the plots are available within this paper and its Supplementary Information. All other relevant data that support the findings of this study are available from the corresponding authors on request. Source data are provided with this paper.

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

## Acknowledgements

S.H. and Z.W. contributed equally to this work. This work was financially supported by the National Natural Science Foundation of China (U1904216, U22A20141). Part of this research was undertaken on the X-ray Absorption Spectroscopy beamline at the Australian Synchrotron, ANSTO. The authors would like to thank Engineer Xiaohui Gu from Hunan Navi New Materials Technology for her assistance with micro-morphology analysis, and Yiyan Testing Technology (Nanjing, China) for their assistance on TOF-SIMS.

## Author contributions

L.C. supervised this work. S.H. and Z.W. conducted the concept design. S.H. and K.L. provided material characterizations. B.J. did the XAS test. S.H. prepared the electrode materials and conducted the experiments. S.H. and P.Q. did the data analysis. S.H. and P.H. drew the schematic. S.H. and Z.W. wrote the paper. X.J., Y.C., and W.W. revised the manuscript. All authors commented on the final manuscript.

## Competing interests

The authors declare no conflict of interest.
