## [Peer Review File · Nature Communications]

nature portfolio

Peer Review FileReviewer comments, first round

Reviewer #1 (Remarks to the Author):

The authors reported a new modified lithium foil obtained by tribochemical process utilizing Zinc dialkyldithiophosphate as the precursor. The characterization results showed that the friction during the extrusion of thin lithium induced reaction between the precursor and lithium generating an interlayer with improved hardness and Young's modulus. Different types of surface characterization suggested an inorganic/organic matrix in the interface layer. The authors also fabricated several lithium-metal battery cells with different cathodes to approve the performance improvement from fresh untreated lithium foil.

In recent years, lithium metal as anode drew much attention as a candidate for next generation of secondary battery component in both academic and industrial communities. Numerous reports with different processes and methodologies on lithium metal anode modifications have been published. Researchers already realized the importance on the interface between the lithium anode and electrolyte. We have seen all kinds of organic and inorganic coatings or interlayer concepts. The idea in this manuscript combining the chemical modification and thin lithium fabrication (extrusion/thickness reduction) is very novel and meaningful, especially from manufacturing aspect.

However, there are several questions and concerns that may need be addressed.

1. Would the authors prepare the thin lithium using the same extrusion process without adding the ZDDP to get similar thickness (e.g. 50um) and provide the characterization for comparison? --- in the manuscript, all comparisons came from fresh lithium, either unprocessed 120um sample, or purchased 50um sample. Yes it is very clear with surface chemical composition differences. However, the authors need to show that the changes on lithium surface came from ZDDP, rather than the extrusion process itself.

2. Would the authors provide all details for the cell fabrication? For example, the "electrode fabrication" section need revised with adequate details for all 3 cathodes used in the experiments. The current version didn't provide the composition of LCO cathode and Sulfur cathode. Also, what is the approximate porosity of the obtained electrodes (or electrode thickness)? Here we suggest at least the authors should provide component composition, porosity/thickness, drying condition.

3. Would the authors provide details for each electrochemical measurement setup. Although in the figure captions there were some descriptions for the C rate, complete testing protocol should be provided – cut-off voltage window, C-rate, definition of C for different chemistry, cell testing condition (temperature).

4. Would the author explain why a variety of electrolytes were used in the work? For example, the authors used 1.0M LiTFSI in DME:DOL=1:1 Vol% with 2.0% LiNO₃ for symmetrical cell tests, however, they switched to 0.6M LiTFSI in DME:DOL=1:1 Vol% with 4.0% LiNO₃ for sulfur cell test. In fact, the first formulation was also widely used for Li-S battery study. So, it is very unclear why the authors switched the electrolyte.

5. Following up with the electrolyte part, another question is for the electrolyte selection for LCO-Li cell. The Localized highly concentrated electrolyte (1.5M LiFSI–DME–TTE electrolyte) was approved to delivery excellent performance on lithium metal anode. However, the authors only showed 50 cycles of the pouch format LCO-Li cell. This is not comparable to other reported results using this LHCE type electrolyte for Li metal batteries. So it is very confusing with the electrolyte selection here. If the authors only want to show the difference between the unmodified lithium and processed lithium with ZDDP, they can use the same carbonate electrolyte that was used for their LFP-Li cell test for LCO-Li cell as well.

6. The authors may want to double check the description in pg7 second paragraph for areal capacity of LCO. It should be ~ 6.5 mAh/cm², rather than 13 mAh/cm² as the cathode was double sided coated.

7. It is not very clear that what cell design was used and showed in the manuscript. The authors mentioned that the LCO-Li cells were with 3 cathodes design in the supporting information, however, the discharge capacity showed in Figure 5a was only ~ 0.4 Ah, which is aligned with single cathode capacity. The authors need make a clear description and Figure captions.

Reviewer #2 (Remarks to the Author):

Comments:

This work introduced zinc dialkyl dithiophosphate (ZDDP) as an anti-pressure, anti-wear additive to engineer bifunctional interfaces on Li metal. The ZDDP additive can trigger tribochemical reaction at ultrathin (5.4-50 μm) Li metal anode, leading to highly stable artificial SEI buildup. A nanoindentation study revealed that surface engineering using Li@ZDDP could enhance the mechanical properties of Li, such as hardness (0.06 \rightarrow 0.84 GPa) and elastic modulus (8.57 \rightarrow 22.67 GPa), resulting in Li dendrite suppression and faster Li⁺ transfer via notable desolvation effect. In addition, DFT calculation was performed to reveal the additional role of organic/inorganic hybrid SEI at Li@ZDDP, which can provide more lithiophilic sites and achieve effective Li-ion desolvation showing better performance at higher current density. The materials and method proposed in this work have originality compared to the previous approaches. The data nicely support the claims that the ultra-high modulus SEI layer provides excellent electrochemical properties at high-current density and for the high volumetric energy density application. Overall, I would like to recommend this manuscript to Nature Communications after revision (major) if some issues and further verification were addressed.

1. The mineral oil used in the roll-pressing process may cause surface contamination. Please provide detailed information if the experiment was well controlled to avoid this concern.
2. TEM and TOF-SIMS analysis were performed only for Li@ZDDP. The comparison with pristine Li should be discussed by collecting the same data set.
3. The authors insisted that the surface nature of Li@ZDDP can tailor the microenvironment of Li⁺ near the surface, facilitating the desolvation (escaping from the solvation shell) and thereby leading to faster Li plating and stripping. Is this possible to provide some experimental evidence by using Raman in the presence of a ZDDP surface?
4. Fundamental explanation is lacking for the description "surface hardness is the key to non-stick rolling." For common knowledge, the sticking phenomena should be related to the chemical affinity between two surfaces. On the other hand, a non-friction surface would be a convincing factor for the non-stick rolling. How does the surface hardness affect the non-sticking phenomena?
5. In the case of a full cell cycling test with the LCO cathode, the data only provided discharge capacity and capacity retention rate of Li@ZDDP. Please provide the data that compares the cycling stability of Li@ZDDP to Li.

Dear Editor and Reviewers,

Thanks for your time to review our work. We are grateful to receive the decision letter and the reviewer's insightful comments regarding our manuscript entitled "Interfacial Friction Enabling Ultrathin Free-Standing Lithium Strips for Stable Lithium Metal Batteries (NCOMMS-22-52280). We have revised the manuscript based on the comments from reviewers, with revisions highlighted in red in the revised manuscript. The point-to-point responses to reviewers' comments are listed below.

We hope that the changes and explanations in this revision are acceptable and look forward to your decision on the publication of our manuscript. If you have any further questions, please do not hesitate to contact us.

Please see the detailed response letter as an attachment.

Thank you for your time.

Sincerely yours,

Libao Chen, Professor

State Key Laboratory of Powder Metallurgy, Central South University

Changsha 410083, Hunan, China

Email: lbchen@csu.edu.cn

COMMENTS TO AUTHOR:

Reviewer #1:

Comments:

The authors reported a new modified lithium foil obtained by tribochemical process utilizing Zinc dialkyldithiophosphate as the precursor. The characterization results showed that the friction during the extrusion of thin lithium induced reaction between the precursor and lithium generating an interlayer with improved hardness and Young's modulus. Different types of surface characterization suggested an inorganic/organic matrix in the interface layer. The authors also fabricated several lithium-metal battery cells with different cathodes to approve the performance improvement from fresh untreated lithium foil.

In recent years, lithium metal as anode drew much attention as a candidate for next generation of secondary battery component in both academic and industrial communities. Numerous reports with different processes and methodologies on lithium metal anode modifications have been published. Researchers already realized the importance on the interface between the lithium anode and electrolyte. We have seen all kinds of organic and inorganic coatings or interlayer concepts. The idea in this manuscript combining the chemical modification and thin lithium fabrication (extrusion/thickness reduction) is very novel and meaningful, especially from manufacturing aspect.

However, there are several questions and concerns that may need be addressed.

1. Would the authors prepare the thin lithium using the same extrusion process without adding the ZDDP to get similar thickness (e.g. 50um) and provide the characterization for comparison? --- in the manuscript, all comparisons came from fresh lithium, either unprocessed 120um sample, or purchased 50um sample. Yes it is very clear with surface chemical composition differences. However, the authors need to show that the changes

on lithium surface came from ZDDP, rather than the extrusion process itself.

Response:

Thank you for your valuable suggestions. By using the same rolling (extrusion) process without adding the ZDDP, only 40- μm broken lithium strips can be prepared due to the low tensile strength and high viscosity (Figure S8a, Supporting Information). Then, one piece of the broken lithium strips was peeled off from the roller and was tested by using the same characterization methods (SIMS-TOF and cryo-TEM) (shown in Figure S3) The results showed that the surface of the broken lithium strip did not have corresponding components concerning Zn, P and S elements. Therefore, the component changes on lithium surface mainly originate from ZDDP, rather than the rolling (extrusion) process itself.

Figure S8. Optical photos of lithium strips (rolling to 40 μm) during actual processing.

(a) No ZDDP was used for rolling

Detailed changes: **Figure S3, Supporting Information**

Figure S3. (a) The TOF-SIMS profiles of different atom counts with the depth increasing on pure Li. (b) 3D structure views for TOF-SIMS depth sputtering on the surface of pure Li. (c) Photographs of the Li surface layer and elemental distribution generated by pure Li cross-sectional observation with cryo-TEM.

Detailed changes: **Paragraph no. 2, Page no. 2**

In contrast, there is no corresponding components concerning Zn, P and S elements on the surface of bare lithium metal (Figure S3, Supporting Information).

2. Would the authors provide all details for the cell fabrication? For example, the “electrode fabrication” section need revised with adequate details for all 3 cathodes used in the experiments. The current version didn’t provide the composition of LCO cathode and Sulfur cathode. Also, what is the approximate porosity of the obtained electrodes (or electrode thickness)? Here we suggest at least the authors should provide component composition, porosity/thickness, drying condition.

Response:

Thank you for your kind suggestion. Details for each cell fabrication, including component composition, porosity/thickness and drying condition, were added in the “Fabrication of Ultra-thin Lithium Anodes” and “Fabrication of Cathodes” sections.

The corresponding descriptions have been supplemented in the revised *Supporting Information*.

Detailed changes: **Supporting Information, Paragraph no. 1, Page no. 2**

Fabrication of Ultra-thin Lithium Anodes. The zinc dialkyldithiophosphate (ZDDP) was purchased from Guangdong Wengjiang Chemical Reagent Co., Ltd. Different thickness (100 μm , 50 μm , 30 μm , 15 μm) of Li was processed by using the lubricants (5 wt% ZDDP in mineral oil) to coat rollers of the mill (MRX-DG150L, Shenzhen Mingruixiang Automation Equipment Co., Ltd) based on 120 μm Li foils (99.9%, China Energy Lithium Co., Ltd, Tianjin). The 50 μm bare Li used in lithium sulfur battery was purchased from China Energy Lithium Co., Ltd in Tianjin. **The rolled lithium anodes with mineral oil (MO, Aladdin) were cleaned by tetrahydrofuran (THF, Dodo Chem) solution then dried in the Ar-filled glovebox.**

Fabrication of Cathodes. *LiCoO₂ Cathodes.* The cathode slurry comprising LiCoO₂, super P and polyvinylidene fluoride (PVDF) (All were purchased from Dongguan Large Electronics Co., Ltd. Guangdong, China) at the mass ratio of 94.9%:2.55%:2.55% were fully dispersed in N-methylpyrrolidone (NMP, Aladdin) for half an hour, then coated onto the 12 μm Al foil and dried overnight under vacuum at 120 °C. LiCoO₂ cathode had an active material loading of 37.7 mg cm⁻² (double sides) and the cathode thickness after rolling was 0.125 mm. ***LiFePO₄ Cathode.*** The cathode slurry comprising LiFePO₄, super P and polyvinylidene fluoride (PVDF) (All were purchased from Dongguan Large Electronics Co., Ltd. Guangdong, China) at the mass ratio of 94.7%:2.65%:2.65% were fully dispersed in N-methylpyrrolidone (NMP, Aladdin) for half an hour, then coated onto the 14 μm C@Al foil and dried overnight under vacuum at 120 °C. LiFePO₄ cathode had an active material loading of 10.0 mg cm⁻² (single side) and the cathode thickness after rolling was 0.085 mm. ***S/C Cathode.*** Sulfur powders (Alfa Aesar) and carbon nanotube (Zhongke era) (mass ratio = 6:4) were ground well with a mortar, then sealed the mixture in an Ar-filled ampoule and placed in a muffle furnace (Hefei Kejing) for 48 hours at 300 °C (S/C). The cathode slurry comprising S/C,

super P and polyvinylidene fluoride (PVDF) at the mass ratio of 8:1:1 was fully dispersed in N-methylpyrrolidone (NMP, Aladdin) for half an hour, then coated onto the 12 μm Al foil and dried overnight under vacuum at 60 °C. S/C cathode had an active material loading of 4.7 mg cm^{-2} (double sides) and the cathode thickness was 0.290 mm.

3. Would the authors provide details for each electrochemical measurement setup. Although in the figure captions there were some descriptions for the C rate, complete testing protocol should be provided – cut-off voltage window, C-rate, definition of C for different chemistry, cell testing condition (temperature).

Response:

We really apologize for the lack of this important information. The corresponding descriptions for each electrochemical measurement setup have been supplemented in the revised Supporting Information.

Detailed changes: **Supporting Information, Paragraph no. 2, Page no. 4**

Electrochemical Measurements. The discharge/charge testing were conducted on a Neware Battery system. All electrochemical tests were performed in a temperature-controlled room at 25°C. All batteries employed Celgard®2400 as the separator. The measurement of the free Li capacity in ultrathin electrodes was done by charging the battery to 1 V at 0.5 mA cm^{-2} with the bare Li as the counter and reference electrode. *Coin cells tests.* CR2016 coin cells were assembled in an Ar-filled glovebox (O_2 and $\text{H}_2\text{O} < 0.5$ ppm). Symmetric coin cells (fresh lithium on each side or Li@ZDDP foil on each side) were assembled with 1.0M LiTFSI in DME:DOL=1:1 Vol% with 2.0% LiNO_3 (LS009, Dodo Chem) as the electrolyte. To standardize the testing, 75 μL of electrolyte was used in each coin cell. Typically, we used a protocol of 1 h of stripping followed by 1 h of plating with a current density of 1.5 mA cm^{-2} to achieve an areal capacity of 1.5 mA h cm^{-2} . The current density for the Li metal plating/stripping was set to 1.5 or 18.0 mA cm^{-2} . Besides, LFP||Li@ZDDP (or Li) full cells were assembled with 1.0M LiPF_6 in EC:EMC:FEC=3:7:1 Vol% (LB515, Dodo Chem) as the electrolyte, which were tested within the voltage range of 2.5-4.2 V. As for cycling tests, the LFP

full cells were activated at 0.1C for 2 cycles firstly, then underwent the charge-discharge test at 1C or 5C ($1C = 170 \text{ mA h g}^{-1}$). As for rate performance tests, the LFP full cells underwent the galvanostatic charge-discharge test in turn of 0.2C, 0.5C, 1C, 2C, 5C, 10C and 0.2C ($1C = 170 \text{ mA h g}^{-1}$). *Pouch cells tests.* The $\text{LiCoO}_2\|\text{Li@ZDDP}$ (or Li) pouch cells and $\text{S}\|\text{Li@ZDDP}$ (or Li) pouch cells were assembled in dry rooms with dew points below -40°C . As for the $\text{LiCoO}_2\|\text{Li@ZDDP}$ (or Li) pouch cell, the size of the Li anode was $50 \text{ mm} \times 80 \text{ mm}$ while the size of the LiCoO_2 cathode was $47 \text{ mm} \times 77 \text{ mm}$. Each electrode had the pole lug at the size of $8 \text{ mm} \times 12 \text{ mm}$. The width of the polypropylene separators was 83 mm. In the layer-by-layer manufacturing process, the $\text{LiCoO}_2\|\text{Li@ZDDP}$ (or Li) pouch cells used three cathodes and four anodes while lithium sulfur pouch cells used two cathodes ($30 \text{ mm} \times 70 \text{ mm}$) and three anodes ($33 \text{ mm} \times 73 \text{ mm}$). Aluminum electrode lugs were welded to the cathodes by an ultrasonic welder while nickel electrode lugs were welded to the anodes by mechanical pressing. $\text{LiCoO}_2\|\text{Li@ZDDP}$ (or Li) pouch cells were tested over the voltage range of 3.0-4.2 V. Each pouch cell was activated at 0.1C for 3 cycles firstly, then underwent the galvanostatic charge-discharge long cycle-lifespan test at 0.5C ($1C = 140 \text{ mAh g}^{-1}$). $\text{S}\|\text{Li@ZDDP}$ (or Li) pouch cells were tested over the voltage range of 1.8-2.5 V. Each pouch cell was tested at 0.2C ($1C = 1000 \text{ mAh g}^{-1}$). The electrolyte/sulfur (E/S) ratio was $3.3 \mu\text{L mg}^{-1}$. Electrochemical impedance spectroscopy (EIS) tests were conducted on a Holland Ivium electrochemical station. The EIS data was recorded in the frequency range from 0.1 to 100000 Hz.

4. Would the author explain why a variety of electrolytes were used in the work? For example, the authors used 1.0M LiTFSI in DME:DOL=1:1 Vol% with 2.0% LiNO₃ for symmetrical cell tests, however, they switched to 0.6M LiTFSI in DME:DOL=1:1 Vol% with 4.0% LiNO₃ for sulfur cell test. In fact, the first formulation was also widely used for Li-S battery study. So, it is very unclear why the authors switched the electrolyte.

Response:

Given that the $\text{Li}\|\text{Li}$ symmetrical cells and the Li-S pouch cells are different battery

systems, we switched the electrolyte from 1.0M LiTFSI in DME:DOL=1:1 Vol% with 2.0% LiNO₃ to 0.6M LiTFSI in DME:DOL=1:1 Vol% with 4.0% LiNO₃. Firstly, increasing the concentration of LiNO₃ can enhance cycling performances in lean-electrolyte Li-S cells. The electrolyte with higher concentration of LiNO₃ facilitates to stabilize the performance of the S cathode in Li-S pouch cells [Adv. Energy Mater. 2020, 10, 2000493]. Due to the presence of the S cathode, we increased the concentration of LiNO₃. At the same time, increasing the concentration of LiNO₃ would decrease the concentration of LiTFSI due to the limited solubility of Li⁺ in DME/DOL solvents. More importantly, 0.5~0.6 M LiTFSI in electrolyte can also help improve the performance of lithium-sulfur pouch cells, as was reported recently [Adv. Mater. 2023, 35, 2208590]. Hence, we used 0.6M LiTFSI in DME:DOL=1:1 Vol% with 4.0% LiNO₃ to optimize the performance of our lean-electrolyte Li-S cells.

5. Following up with the electrolyte part, another question is for the electrolyte selection for LCO-Li cell. The Localized highly concentrated electrolyte (1.5M LiFSI-DME-TTE electrolyte) was approved to delivery excellent performance on lithium metal anode. However, the authors only showed 50 cycles of the pouch format LCO-Li cell. This is not comparable to other reported results using this LHCE type electrolyte for Li metal batteries. So it is very confusing with the electrolyte selection here. If the authors only want to show the difference between the unmodified lithium and processed lithium with ZDDP, they can use the same carbonate electrolyte that was used for their LFP-Li cell test for LCO-Li cell as well.

Response:

Thanks for your valuable suggestion. According to your suggestion, we re-verified the performance of the LCO pouch cell here, which used the same electrolyte with LFP||Li cell (shown in Figure 5a). We also complemented the electrochemical performance of pure lithium for comparing. In fact, due to the thin negative electrode (15 μm), the pouch cell using a localized highly concentrated electrolyte was not as good as other reported results. The corresponding descriptions have been corrected in the revised

manuscript.

Detailed changes: **Figure 5a**

Figure 5. (a) Cycling performance of the pouch cell with LiCoO₂||Li@ZDDP (15 μm) using 1.0M LiPF₆ in EC:EMC:FEC=3:7:1 Vol% electrolytes at 0.5C.

Detailed changes: **Paragraph no. 2, Page no. 7**

To verify the practical application of the prepared ultra-thin Li@ZDDP anode, full cells of S||Li@ZDDP, LFP||Li@ZDDP, LCO||Li@ZDDP were assembled, respectively.

Figure 5a shows the cycling performance of the pouch cell using 15 μm Li@ZDDP anodes and commercial LiCoO₂ cathodes with a double-sided load of 37.7 mg cm⁻² to evaluate its application performance. Its theoretical discharge capacity is about 0.56 Ah (~ 2.6 mA h cm⁻²). The capacity retention of LCO||Li@ZDDP remains ~100% after 50 cycles, while the capacity retention of LCO||Li keeps only 60% after 50 cycles, revealing that the Li@ZDDP anode exhibits better electrochemical stability.

6. The authors may want to double check the description in pg7 second paragraph for areal capacity of LCO. It should be ~ 6.5 mAh/cm², rather than 13 mAh/cm² as the cathode was double sided coated.

Response:

Thanks for your careful review. Here we make a mistake and now make corrections.

As we have assembled new pouch batteries using new carbonate electrolyte and new LiCoO₂ cathodes (with a double-sided load of 37.7 mg cm⁻²), the areal capacity of LCO in this revised version was calculated as follows,

$$37.7 \text{ mg cm}^{-2} \times 140 \text{ mA h g}^{-1} \div 2 \text{ (double sided coated)} = 2.6 \text{ mA h cm}^{-2}$$

The corresponding descriptions have been corrected in the revised manuscript.

Optical images of new LiCoO₂ cathodes

Detailed changes: **Paragraph no. 2, Page no. 7**

Figure 5a shows the cycling performance of the pouch cell using 15 μm Li@ZDDP anodes and commercial LiCoO₂ cathodes with a double-sided load of 37.7 mg cm⁻² to evaluate its application performance. Its theoretical discharge capacity is about 0.56 A h (~ 2.6 mA h cm⁻²). The capacity retention of LCO||Li@ZDDP remains ~100% after 50 cycles, while the capacity retention of LCO||Li keeps only 60% after 50 cycles, revealing that the Li@ZDDP anode exhibits better electrochemical stability.

7. It is not very clear that what cell design was used and showed in the manuscript. The authors mentioned that the LCO-Li cells were with 3 cathodes design in the supporting information, however, the discharge capacity showed in Figure 5a was only ~ 0.4 Ah, which is aligned with single cathode capacity. The authors need make a clear description and Figure captions.

Response:

Thank you for pointing this out. Based on the revision of *Question 6*, the specific calculation process for the discharge capacity was described as follows,

$$2.6 \text{ mA h cm}^{-2} \times 4.7 \text{ cm} \times 7.7 \text{ cm} \times 2 \text{ (double sided coated)} \times 3 = 0.56 \text{ A h}$$

The corresponding descriptions have been corrected in the revised manuscript.

Detailed changes: **Paragraph no. 2, Page no. 7**

Figure 5a shows the cycling performance of the pouch cell using 15 μm Li@ZDDP anodes and commercial LiCoO₂ cathodes with a double-sided load of 37.7 mg cm⁻² to evaluate its application performance. Its theoretical discharge capacity is about 0.56 A h (~ 2.6 mA h cm⁻²). The capacity retention of LCO||Li@ZDDP remains ~100% after 50 cycles, while the capacity retention of LCO||Li keeps only 60% after 50 cycles, revealing that the Li@ZDDP anode exhibits better electrochemical stability.

Reviewer #2:

Comments:

This work introduced zinc dialkyl dithiophosphate (ZDDP) as an anti-pressure, anti-wear additive to engineer bifunctional interfaces on Li metal. The ZDDP additive can trigger tribochemical reaction at ultrathin (5.4-50 μm) Li metal anode, leading to highly stable artificial SEI buildup. A nanoindentation study revealed that surface engineering using Li@ZDDP could enhance the mechanical properties of Li, such as hardness (0.06 \rightarrow 0.84 GPa) and elastic modulus (8.57 \rightarrow 22.67 GPa), resulting in Li dendrite suppression and faster Li⁺ transfer via notable desolvation effect. In addition, DFT calculation was performed to reveal the additional role of organic/inorganic hybrid SEI at Li@ZDDP, which can provide more lithiophilic sites and achieve effective Li-ion desolvation showing better performance at higher current density. The materials and method proposed in this work have originality compared to the previous approaches. The data nicely support the claims that the ultra-high modulus SEI layer provides excellent electrochemical properties at high-current density and for the high volumetric energy density application.

Overall, I would like to recommend this manuscript to Nature Communications after revision (major) if some issues and further verification were addressed.

1. The mineral oil used in the roll-pressing process may cause surface contamination. Please provide detailed information if the experiment was well controlled to avoid this concern.

Response:

We are grateful for this important comment. As tetrahydrofuran (THF) and mineral oil (MO) are mutually soluble (see photos below), the lithium anodes were rinsed with tetrahydrofuran (THF) solvents after rolling. Then, they were dried in the Ar-filled glovebox with the volatilization of THF. It is noted that THF is highly volatile and does not react with lithium metal [*Small* 2021, 17, 2103375]. Therefore, the mineral oil used

in the roll-pressing process can be removed from lithium strips without causing surface contamination. This method was also supplemented in the *Supporting Information*.

(a) The MO liquid. (b) Adding the THF into the MO.

Detailed changes: **Supporting Information, Paragraph no. 1, Page no. 2**

Fabrication of Ultra-thin Lithium Anodes. The zinc dialkyldithiophosphate (ZDDP) was purchased from Guangdong Wengjiang Chemical Reagent Co., Ltd. Different thickness (100 μm , 50 μm , 30 μm , 15 μm) of Li was processed by using the lubricants (5 wt% ZDDP in mineral oil) to coat rollers of the mill (MRX-DG150L, Shenzhen Mingruixiang Automation Equipment Co., Ltd) based on 120 μm Li foils (99.9%, China Energy Lithium Co., Ltd, Tianjin). The 50 μm bare Li used in lithium sulfur battery was purchased from China Energy Lithium Co., Ltd in Tianjin. The rolled lithium anodes with mineral oil (MO, Aladdin) were cleaned by tetrahydrofuran (THF, Dodo Chem) solution then dried in the Ar-filled glovebox.

2. TEM and TOF-SIMS analysis were performed only for Li@ZDDP. The comparison with pristine Li should be discussed by collecting the same data set.

Response:

Thanks for your suggestions. Compared with Li@ZDDP, the same characterization methods (SIMS-TOF and cryo-TEM) were used to analysis the surface of the pure lithium (shown in **Figure S3**). The results showed that the surface of bare lithium without adding the ZDDP did not have corresponding components concerning Zn, P and S elements.

Detailed changes: **Figure S3, Supporting Information**

Figure S3. (a) The TOF-SIMS profiles of different atom counts with the depth increasing on pure Li. (b) 3D structure views for TOF-SIMS depth sputtering on the surface of pure Li. (c) Photographs of the Li surface layer and elemental distribution generated by pure Li cross-sectional observation with cryo-TEM.

Detailed changes: **Paragraph no. 2, Page no. 2**

In contrast, there is no corresponding components concerning Zn, P and S elements on the surface of bare lithium metal (Figure S3, Supporting Information).

3. The authors insisted that the surface nature of Li@ZDDP can tailor the microenvironment of Li⁺ near the surface, facilitating the desolvation (escaping from the solvation shell) and thereby leading to faster Li plating and stripping. Is this possible to provide some experimental evidence by using Raman in the presence of a ZDDP surface?

Response:

Thank you for your valuable suggestion. *In situ* Raman tests were conducted to provide the evidence of the desolvation effect at the Li@ZDDP interface. It shows that weak solvated structures of Li⁺ occur near the Li@ZDDP interface, as the peak of S-N-S

underwent blueshifts with the progress of electroplating. In contrast, the degree of solvation is fairly high near the pure lithium interface, as the peak of S-N-S of the pure lithium anode almost disappeared. In summary, the strong desolvation effect near the Li@ZDDP interface facilitates the fast electrodeposition [Adv. Mater. 2023, 35, 2208590].

Detailed changes: **Paragraph no. 1, Page no. 6**

To investigate the Li⁺ solvation structure near the Li or Li@ZDDP interface, we conducted Raman spectral tests using 1.0 M LiTFSI in DME:DOL=1:1 Vol% electrolyte. Figure S17 a-c (supporting information) shows Raman spectra results, where the S–N–S bending peak of the TFSI⁻ at 736.80 cm⁻¹ (near the Li@ZDDP interface) undergoes redshift (vs. 755.10 cm⁻¹ in electrolyte) when dissolved in DOL/DME solvents. While the S-N-S peak near the Li interface almost disappeared (Figure S17e, supporting information). This implies a high degree of solvation near the pure lithium interface. On the contrary, the S-N-S peak near the Li@ZDDP interface underwent blueshift with the progress of electroplating (Figure S17d, supporting information). This indicates that there is much stronger cation-anion interaction near the Li@ZDDP interface and the desolvation effect is significant herein.⁴⁷

Detailed changes: **Figure S17, Supporting Information**

Figure S17. *In situ* Raman spectra of electrolyte near anode–electrolyte interface

during Li@ZDDP (a, d) and Li plating (b, e) with 1.0 M LiTFSI in DME:DOL=1:1 Vol% at the plating current density of 3 mA cm⁻². (c) Initial Raman spectra of Li and Li@ZDDP near anode–electrolyte interface.

Detailed changes: **Supporting Information, Paragraph no. 2, Page no. 3**

***In situ* Raman Characterization.** Raman spectra tests were conducted with a Horiba Jobin Y von HR Evolution Raman spectrometer under a Raman laser wavenumber of 532 nm from a neodymium-doped yttrium aluminum garnet (Nd:YAG) laser operating at 120 mW. The Li deposition in Raman spectroscopic cell (K007, Tianjin Aida Hengsheng Technology Development Co., Ltd) with 1.0 M LiTFSI in DME:DOL=1:1 Vol% was performed at 3 mA cm⁻² lasting 20 min which implies a theoretical deposition thickness of ~5 μm, while the laser spot has a diameter of ~1.5 μm. For the *in situ* Raman test, the distance between the laser spot and the substrate was set as 6.5 μm.

4. Fundamental explanation is lacking for the description “surface hardness is the key to non-stick rolling.” For common knowledge, the sticking phenomena should be related to the chemical affinity between two surfaces. On the other hand, a non-friction surface would be a convincing factor for the non-stick rolling. How does the surface hardness affect the non-sticking phenomena?

Response:

Thank you for your insightful comments. In fact, the hardened interface significantly changes the surface physicochemical properties and can reduce the interfacial viscosity on the lithium strip. What’s more, separating the lithium strip from the roller requires overcoming the liquid-phase adsorption from the oil between the lithium strip and the roller. Due to the relatively low tensile strength of lithium, it is damaged easily without surface hardening when separating. On the other hand, the roller sticking phenomenon occurs as the adhesion wear between the Li surface and the roller with incomplete surface separation [*Frontiers in Chemistry* 2022, 10, 852371; *Physical Review E* 2020, 102, 043001]. What we explain here is that as the surface hardness increases, the adhesive wear degree of lithium strips is much smaller, which is conducive to the

complete separation of the interface between the lithium strips and the roller. Since it is not clearly stated in the manuscript, we have revised it accordingly. The corresponding descriptions have been corrected in the revised manuscript.

Detailed changes: **Paragraph no. 3, Page no. 3**

This process layer formed by the ZDDP reaction has excellent mechanical properties, and the surface layer is mechanically tested by the nanoindentation measurements in continuous multi-cycles loading mode (schematic diagram shown in Figure 3a). The results show that the surface Young's modulus of the processed lithium metal (Li@ZDDP) is about 25.90 GPa, which is about 3.5 times higher than the lithium strip (7.43 GPa, Figure 3b). Crucially, the surface hardness is increased to 0.84 GPa (vs. 0.06 GPa of Li metal matrix), almost increasing to 15 times as compared with that of Li (Figure 3c). **Due to the relatively low tensile strength of lithium, it is damaged easily without surface hardening when separating. The high surface hardness is conducive to the separation of the interface completely between the lithium strips and the roller.**

5. In the case of a full cell cycling test with the LCO cathode, the data only provided discharge capacity and capacity retention rate of Li@ZDDP. Please provide the data that compares the cycling stability of Li@ZDDP to Li.

Response:

Thank you for your suggestions. Here we re-evaluated the performance of the LCO pouch cell, which used the same electrolyte with LFP||Li cell (shown in Figure 5a). The cycling stability of the LCO||Li pouch cell and the LCO||Li@ZDDP pouch cell were compared accordingly. The capacity retention of LCO||Li@ZDDP remains ~100% after 50 cycles, while the capacity retention of LCO||Li keeps only 60% after 50 cycles, revealing that the Li@ZDDP anode exhibits better electrochemical stability. The corresponding descriptions have been corrected in the revised manuscript.

Detailed changes: **Figure 5a**

Figure 5. (a) Cycling performance of the pouch cell with $\text{LiCoO}_2||\text{Li@ZDDP}$ ($15 \mu\text{m}$) using 1.0M LiPF_6 in $\text{EC}:\text{EMC}:\text{FEC}=3:7:1$ Vol% electrolytes at 0.5C .

Detailed changes: Paragraph no. 2, Page no. 7

To verify the practical application of the prepared ultra-thin Li@ZDDP anode, full cells of $\text{S}||\text{Li@ZDDP}$, $\text{LFP}||\text{Li@ZDDP}$, $\text{LCO}||\text{Li@ZDDP}$ were assembled, respectively.

Figure 5a shows the cycling performance of the pouch cell using $15 \mu\text{m Li@ZDDP}$ anodes and commercial LiCoO_2 cathodes with a double-sided load of 37.7 mg cm^{-2} to evaluate its application performance. Its theoretical discharge capacity is about 0.56 A h ($\sim 2.6 \text{ mA h cm}^{-2}$). The capacity retention of $\text{LCO}||\text{Li@ZDDP}$ remains $\sim 100\%$ after 50 cycles, while the capacity retention of $\text{LCO}||\text{Li}$ keeps only 60% after 50 cycles, revealing that the Li@ZDDP anode exhibits better electrochemical stability.

Detailed changes: Supporting Information, Paragraph no. 1, Page no. 2

Fabrication of Cathodes. *LiCoO₂ Cathodes.* The cathode slurry comprising LiCoO_2 , super P and polyvinylidene fluoride (PVDF) (All were purchased from Dongguan Large Electronics Co., Ltd. Guangdong, China) at the mass ratio of $94.9\%:2.55\%:2.55\%$ were fully dispersed in N-methylpyrrolidone (NMP, Aladdin) for half an hour, then coated onto the $12 \mu\text{m}$ Al foil and dried overnight under vacuum at $120 \text{ }^\circ\text{C}$. LiCoO_2 cathode had an active material loading of 37.7 mg cm^{-2} (double sides) and the cathode thickness after rolling was 0.125 mm .

Electrochemical Measurements. The discharge/charge testing were conducted on a Neware Battery system. All electrochemical tests were performed in a temperature-controlled room at 25°C. All batteries employed Celgard®2400 as the separator. The measurement of the free Li capacity in ultrathin electrodes was done by charging the battery to 1 V at 0.5 mA cm⁻² with the bare Li as the counter and reference electrode.

Coin cells tests. CR2016 coin cells were assembled in an Ar-filled glovebox (O₂ and H₂O < 0.5 ppm). Symmetric coin cells (fresh lithium on each side or Li@ZDDP foil on each side) were assembled with 1.0M LiTFSI in DME:DOL=1:1 Vol% with 2.0% LiNO₃ (LS009, Dodo Chem) as the electrolyte. To standardize the testing, 75 μL of electrolyte was used in each coin cell. Typically, we used a protocol of 1 h of stripping followed by 1 h of plating with a current density of 1.5 mA cm⁻² to achieve an areal capacity of 1.5 mA h cm⁻². The current density for the Li metal plating/stripping was set to 1.5 or 18.0 mA cm⁻². Besides, LFP||Li@ZDDP (or Li) full cells were assembled with 1.0M LiPF₆ in EC:EMC:FEC=3:7:1 Vol% (LB515, Dodo Chem) as the electrolyte, which were tested within the voltage range of 2.5-4.2 V. As for cycling tests, the LFP full cells were activated at 0.1C for 2 cycles firstly, then underwent the charge-discharge test at 1C or 5C (1C = 170 mA h g⁻¹). As for rate performance tests, the LFP full cells underwent the galvanostatic charge-discharge test in turn of 0.2C, 0.5C, 1C, 2C, 5C, 10C and 0.2C (1C = 170 mA h g⁻¹).

Pouch cells tests. The LiCoO₂||Li@ZDDP (or Li) pouch cells and S|| Li@ZDDP (or Li) pouch cells were assembled in dry rooms with dew points below - 40°C. As for the LiCoO₂||Li@ZDDP (or Li) pouch cell, the size of the Li anode was 50 mm × 80 mm while the size of the LiCoO₂ cathode was 47 mm × 77 mm. Each electrode had the pole lug at the size of 8 mm × 12 mm. The width of the polypropylene separators was 83 mm. In the layer-by-layer manufacturing process, the LiCoO₂||Li@ZDDP (or Li) pouch cells used three cathodes and four anodes while lithium sulfur pouch cells used two cathodes (30 mm × 70 mm) and three anodes (33 mm × 73 mm). Aluminum electrode lugs were welded to the cathodes by an ultrasonic welder while nickel electrode lugs were welded to the anodes by mechanical pressing.

LiCoO₂||Li@ZDDP (or Li) pouch cells were tested over the voltage range of 3.0-4.2 V. Each pouch cell was activated at 0.1C for 3 cycles firstly, then underwent the galvanostatic charge-discharge long cycle-lifespan test at 0.5C (1C = 140 mAh g⁻¹).

Reviewer comments, second round

Reviewer #1 (Remarks to the Author):

The authors provided a thorough revision of the manuscript based on previous reviewer's comments. I think all of the questions were clearly addressed with additional data, analysis, and discussion. Therefore, I think the current version meets the standard of publication on Nature Communications journal.

Reviewer #2 (Remarks to the Author):

Additional experiments and interpretations regarding the question and comment from the reviewer have been answered. However, the author's interpretation of Raman analysis to support the role of ZDDP in facilitating desolvation seems to have some errors compared to the reference the author provided. In addition, the answer about surface hardness doesn't seem valid. Therefore, if this issue can be revised once again, I would like to recommend this manuscript to Nature Communication.

1. The author mentioned that THF does "not" react with Li metal; therefore used THF to rinse mineral oil from the Li metal surface after the rolling process.

2. Raman data interpretation does not logically fit. The peak existing at 736.8 cm^{-1} is known as a free TFSI⁻ and 755.1 is a peak of TFSI⁻ that is being coordinated. When the peak intensity of 736.8 cm^{-1} is higher, it can be said that the ratio of free TFSI⁻ is larger, which means that the desolvation energy is higher. Therefore, looking at Figure S17 c, it can be confirmed that the intensity of the free TFSI⁻ the peak is higher than that of Li at Li@ZDDP-0 min, which may conflict with the author's claim that ZDDP can facilitate the desolvation and lead to fast Li plating and stripping. Also, there are no interpretations in Figure S17 a and b.

3. For Fundamental explanation is lacking for the description "surface hardness is the key to non-stick rolling". It doesn't seem to explain exactly what relationship surface hardness, adhesive wear degree, and tensile strength have in the Revised answer. Also, if rolling is carried out using only mineral oil without ZDDP, in my opinion, it seems that Li will not adhere to the roll even if hardness does not occur. How can these phenomena be explained?Li.

Reviewer #2:

Comments:

Additional experiments and interpretations regarding the question and comment from the reviewer have been answered. However, the author's interpretation of Raman analysis to support the role of ZDDP in facilitating desolvation seems to have some errors compared to the reference the author provided. In addition, the answer about surface hardness doesn't seem valid. Therefore, if this issue can be revised once again, I would like to recommend this manuscript to Nature Communication.

1. The author mentioned that THF does "not" react with Li metal; therefore, used THF to rinse mineral oil from the Li metal surface after the rolling process.

Response:

We are grateful for your important comments. As it was widely reported, that THF can be used for cleaning the surface of lithium metals after modifying [*Nat. Energy* 2017, 2, 17119; *Adv. Funct. Mater.* 2020, 2002414; *Chinese Chem. Lett.* 2023, 1001-8417, 108151; *Adv. Mater.* 2020, 32, 1902724]. Meanwhile, THF can also be used as a good inert solvent for studying the reaction between other reagents and lithium metal [*Adv. Mater.* 2018, 30, 36, 1705711; *J. Am. Chem. Soc.* 1958, 80, 2, 380–383; *J. Phys. Org. Chem.*, 2003 16: 669-674]. Therefore, the literatures above reckon that THF is relatively inert to Li metal, and it can be used to clean the lithium surface as it is also mutually soluble with the mineral oil.

2. Raman data interpretation does not logically fit. The peak existing at 736.8 cm^{-1} is known as a free TFSI- and 755.1 is a peak of TFSI- that is being coordinated. When the peak intensity of 736.8 cm^{-1} is higher, it can be said that the ratio of free TFSI- is larger, which means that the desolvation energy is higher. Therefore, looking at Figure S17 c, it can be confirmed that the intensity of the free TFSI- the peak is higher than that of Li

at Li@ZDDP-0 min, which may conflict with the author's claim that ZDDP can facilitate the desolvation and lead to fast Li plating and stripping. Also, there are no interpretations in Figure S17 a and b.

Response:

Thanks for your insightful comments. We believe that the *in situ* Raman results are rational herein, and it does not conflict with the conclusions that Li@ZDDP can facilitate the desolvation and lead to fast Li plating and stripping. Firstly, the desolvation energy should be correlated to the Raman peak position, rather than the intensity of the peak [*J. Am. Chem. Soc.* 2017, 139, 51; *Energy Storage Mater.* 2022, 52, 69-75]. Secondly, the intense peak of Li@ZDDP represents a higher ratio of free TFSI⁻ at the Li@ZDDP interface due to the strong absorption ability of the Li@ZDDP interface. Thirdly, as the electroplating goes, the S–N–S bending peak of the TFSI⁻ at 736.80 cm⁻¹ (near the Li@ZDDP interface) undergoes blueshift, which indicates that the cation-anion clusters near the surface of Li@ZDDP transform to contact ion pair (CIP) and aggregates (AGG) states, as was shown in Figure 4f. When electroplating for 16 min, the peak of Li@ZDDP-14 min (point 8.0 shown in Figure S17d, Supporting Information) is located at about 747.0 cm⁻¹ while Li-14 min (point 8.0 shown in Figure S17e, Supporting Information) remains no distinct peak corresponding to free anion state. Therefore, Li@ZDDP can strengthen the desolvation effect when electroplating going.

We are sorry for causing misunderstanding due to the lack of necessary description for the Raman data. Besides, the interpretations of Figure S17a and S17b have also been supplemented in the manuscript as was shown below,

Figure S17a and S17b (Supporting Information) presents the *in situ* Raman data of Li@ZDDP and Li as a 2D contour plot with intensity in color covering a selected region between 600 and 1050 cm⁻¹, respectively, where the TFSI⁻ peaks are framed. According to the principle of local "electrical neutrality", the dynamic change of TFSI⁻ during electrodeposition can reflect the change of the concentration of Li⁺. With the increase of the electroplating time, the concentration of Li⁺ near the interface on Li is rapidly

depleted due to the dendrite growth while the concentration of Li^+ has no obvious change near the Li@ZDDP due to the fast adsorption of the organic/inorganic hybrid interface and the continuous uniform deposition.

Detailed changes: **Figure S17, Supporting Information**

Figure S17. *In situ* Raman spectra of electrolyte near anode–electrolyte interface during Li@ZDDP (a, d) and Li plating (b, e) with 1.0 M LiTFSI in $\text{DME:DOL}=1:1$ Vol% at the plating current density of 3 mA cm^{-2} . (c) Initial Raman spectra of Li and Li@ZDDP near anode–electrolyte interface.

Detailed changes: **Paragraph no. 1, Page no. 6**

The initial Raman spectra results of bare Li and Li@ZDDP near the interface are shown in the Figure S17c (Supporting Information). The cation-anion clusters of Li@ZDDP -0 min ($\sim 736.8 \text{ cm}^{-1}$) assigned as free anion (FA) is different with that of LiTFSI ($\sim 755.1 \text{ cm}^{-1}$). Meanwhile, the cation-anion clusters of Li -0 min assigned as FA as well. With prolonged electroplating time, the S–N–S bending peak of the TFSI- at 736.80 cm^{-1} (near the Li@ZDDP interface) undergoes blueshift, which indicates that the cation-anion clusters near the surface of Li@ZDDP transform to contact ion pair (CIP) and aggregates (AGG) states shown in Figure 4f.⁴⁸ The peak of Li@ZDDP -14 min (point 8.0, 0 ~ 14 min labeled point 0.0 ~ 8.0) shown in Figure S17d (Supporting Information) is about 747.0 cm^{-1} while Li -14 min (point 8.0, shown in Figure S17e, Supporting

Information) still shows no distinct peak corresponding to free anion state. This indicates that Li@ZDDP can realize the desolvation effect when electroplating. Figure S17a and S17b (Supporting Information) presents the *in situ* Raman data of Li@ZDDP and Li as a 2D contour plot with intensity in color covering a selected region between 600 and 1050 cm^{-1} , respectively, where the TFSI⁻ peaks are framed. According to the principle of local "electrical neutrality", the dynamic change of TFSI⁻ during electrodeposition can reflect the change of the concentration of Li⁺. With the increase of the electroplating time, the concentration of Li⁺ near the interface on Li is rapidly depleted due to the dendrite growth while the concentration of Li⁺ has no obvious change near the Li@ZDDP due to the fast adsorption of the organic/inorganic hybrid interface and the continuous uniform deposition.⁴⁹

3. For Fundamental explanation is lacking for the description "surface hardness is the key to non-stick rolling". It doesn't seem to explain exactly what relationship surface hardness, adhesive wear degree, and tensile strength have in the Revised answer. Also, if rolling is carried out using only mineral oil without ZDDP, in my opinion, it seems that Li will not adhere to the roll even if hardness does not occur. How can these phenomena be explained?

Response:

Thank you for your valuable comments. We realize that the expression of "surface hardness is the key to non-stick rolling" is not that inaccurate. As A. Almqvist *et al.* reported that the average interfacial separation is closely related to the pressure and the fractional contact area [*J. Mech. Phys. Solids* 2011, 59, 2355–2369]. The sticking phenomena should be related to the applied load and the chemical affinity between two surfaces. When exerting high load, it will be difficult to separate ultrathin lithium strips completely from the roller due to the low tensile strength of pure ultrathin lithium strips. In contrast, Li@ZDDP can withstand higher tensile stress to separate the ultrathin Li matrix from the roller as ZDDP can form robust tribofilms on the strip surface with increased surface hardness [*Adv. Mater.* 2015, 27, 4767-4774].

On one hand, hardness is a typical factor reflecting the mechanical characteristics of

the material microstructures [Additive Manufacturing 2020, 35, 101282]. According to the Archard wear equation, the volume of material loss (V) during friction is directly proportional to the applied load, W , and the sliding distance, s , and inversely proportional to the hardness of the material, H :

$$V = K / W s H$$

where K is a wear coefficient [Wear 2017, 309-391, 236-245]. For ultra-thin lithium strips, the adhesion wear rate can be significantly reduced with increased surface hardness. On the other hand, tensile strength is an important factor for ultra-thin lithium strips to pull the lithium strip forward, thus achieving continuous production during the rolling process. The lithium strip would rupture when the tensile strength is too low. As lithium strips become thinner, the harden modified surface would greatly improve the tensile strength due to the increased proportion of the surface layer relatively to the Li matrix.

In summary, increasing the surface hardness can effectively reduce the adhesion wear and facilitate the complete separation between lithium strips and the roller. As the thickness decreases, increased surface hardness can effectively improve the tensile strength as well.

As for the absence of ZDDP, broken lithium strips will be obtained when the thickness reduces to 40- μm in the experiment (Figure S8a, Supporting Information). As mineral oil was used alone, the phenomenon of sticking roller would still occur, owing to the rupture of the oil film under heavy load [Pet. Chem. 2016, 56, 879–882]. In contrast, ZDDP can be involved in the formation of tribofilms as a reactant that increases the surface hardness, which can withstand higher tensile stress to separate the pure Li matrix from the roller [Adv. Mater. 2015, 27, 4767-4774]. As a result, the rolling stick phenomenon can be eliminated with the aid of ZDDP during the rolling process.

Figure S8. Optical photos of lithium strips (rolling to 40 μm) during actual processing. (a) No ZDDP was used for rolling

We have modified the relevant description further in the manuscript.

Detailed changes: **Paragraph no. 3, Page no. 3**

For ultra-thin lithium strips, the adhesion wear rate can be significantly reduced with increased surface hardness.³⁸ As lithium strips become thinner, the ZDDP-derived harden surface would greatly improve the tensile strength due to the increased proportion of the surface layer relatively to the Li matrix. In summary, the high surface hardness of the tribofilm is conducive to separate the interface completely between the lithium strips and the roller when rolling toward ultrathin scale.

Reviewer comments, third round

Reviewer #2 (Remarks to the Author):

Additional explanation regarding the questions and comments from the reviewer was successfully provided. The author carefully explained how ZDDP can facilitate desolvation step-by-step. In addition, the author provided basic concepts and information about the relationship between hardness, adhesive wear degree, and tensile strength to explain the surface hardness. Therefore, I would like to recommend this manuscript to Nature Communication.